# Effect of climate on traits of dominant and rare tree species in the world's forests

Species' traits and environmental conditions determine the abundance of tree species across the globe. The extent to which traits of dominant and rare tree species differ remains untested across a broad environmental range, limiting our understanding of how species traits and the environment shape forest functional composition. We use a global dataset of tree composition of >22,000 forest plots and 11 traits of 1663 tree species to ask how locally dominant and rare species differ in their trait values, and how these differences are driven by climatic gradients in temperature and water availability in forest biomes across the globe. We find three consistent trait differences between locally dominant and rare species across all biomes; dominant species are taller, have softer wood and higher loading on the multivariate stem strategy axis (related to narrow tracheids and thick bark). The difference between traits of dominant and rare species is more strongly driven by temperature compared to water availability, as temperature might affect a larger number of traits. Therefore, climate change driven global temperature rise may have a strong effect on trait differences between dominant and rare tree species and may lead to changes in species abundances and therefore strong community reassembly.

Plant communities typically consist of a relatively few dominant and many rare species (MacArthur, 1957; Preston, 1948). Dominant and rare species both contribute to ecosystem function: dominant species provide the majority of ecosystem services, and rare species can increase ecosystem multifunctionality by expanding trait diversity[1–4]. Species traits in combination with abiotic and biotic environmental conditions therefore drive the relative abundance of species in local communities[5]. Macroclimate is an important abiotic trait filter that determines the global distribution of forest biomes[6,7] and tree species[8,9]. Climate change will therefore have a strong effect on the occurrence and distribution of forest biomes, traits, and consequently, forest ecosystem functioning[10–12]. However, the extent to which individual traits of locally dominant and rare tree species differ, and how these differences are affected by climate, remains largely unexplored at a global scale. This lack of knowledge limits our understanding on the processes determining species abundances, functional significance of dominant and rare tree species across the globe and how this is affected by climate[3,13].

Community assembly is the process by which species are filtered out from the regional species pool into the local community based on their functional traits, ecological niches or stochastic processes. In this process, climatic factors such as temperature and precipitation, as well as biotic factors such as facilitation, competition, herbivory and pathogens act as filters on species membership in particular assemblages[14,15]. It is suggested that the strength of different filters depend on the environment, with stronger abiotic filtering at higher latitudes because of harsh environmental conditions and stronger biotic filtering at lower latitudes, because of intense competition under productive conditions[16].

After a species' establishment, its abundance is defined besides habitat suitability by competitive ability related to species' traits[5,17]. A trait is defined as any morphological, physiological or phenological feature measurable at the individual plant level that affects plant performance[18]. In forests across the globe, high wood density and low specific leaf area (SLA) are associated with a stronger competitive ability[19]. Higher wood density generally increases tissue longevity and

✉ e-mail: irishordijk@hotmail.com

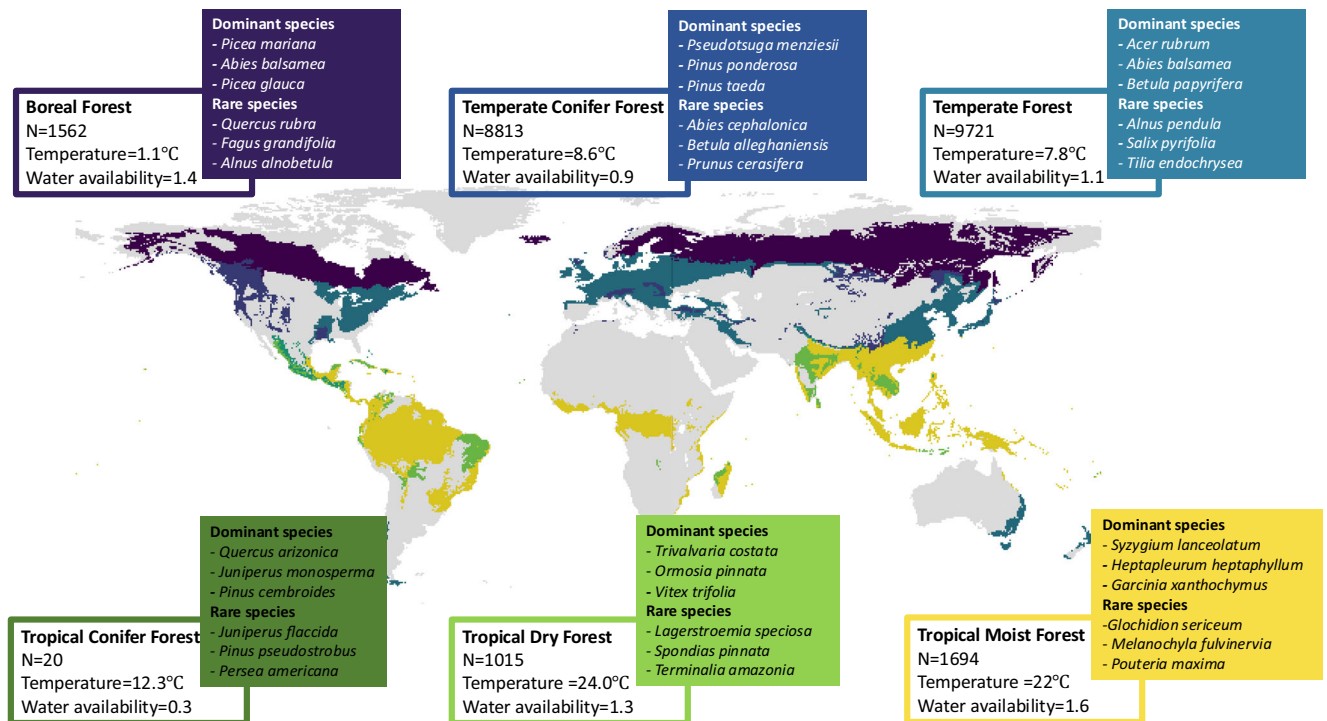

**Fig. 1 | The six forest biomes, with the number of plots, mean annual temperature and mean water availability index displayed.** The three most common dominant and randomly chosen rare tree species, according to our dataset, are indicated per forest biome. An overview of the temperature and water availability range per biome can be found in Fig. S5.

shade-tolerance, as it allows species to make persistent deeper and wider crowns that enhance light interception and shade out other species[20–22]. SLA reflects the life-history strategy of plants, with a high SLA associated with a short leaf lifespan and high growth rate, and a lower SLA associated with a long leaf lifespan and efficient nutrient conservation[16,23,24].

Plant trait occurrence and abundance is affected by different factors, among which temperature and water availability are of primary importance[10,25]. Temperature affects the energy balance of leaves, and therefore the balance between photosynthetic rate and respiration[26,27]. This leads to hump-shaped photosynthesis-temperature response curves which may have profound effects on whole-plant growth rate[28]. Lower latitudes and altitudes have a higher mean annual temperature and are associated with more productive environments (when water and nutrients are not limiting), taller trees, increased construction costs of stem and leaves (high wood density and low SLA that enhance tissue longevity and shade tolerance), and increased seed mass (provide seedlings with sufficient reserves to establish and survive in the shade)[24,25,29,30]. Climatic plant water availability, expressed as aridity (i.e., the ratio of mean annual precipitation over mean annual evapotranspiration), is mainly related to traits connected to drought tolerance (high wood density and cavitation resistance), drought avoidance (deciduous leaves with high SLA and deep roots), or efficient water use (wide vessels, high leaf nitrogen concentration and photosynthetic water use efficiency)[25,31,32]. Extreme temperatures in combination with drought can exacerbate water stress, damage plant tissues, and ultimately lead to plant mortality and species exclusion[33–35].

In addition to maladaptation to the macroclimate and competition, tree species can be locally rare because of metapopulation processes (e.g., recent invaders or in the process of local extinction)[36], extinction of mutualists (e.g., pollinators, dispersers)[37], (biogeographical) historical legacies[38–40], habitat specialization (e.g., specialized for locally rare habitats such as streams, rocky outcrops or treefall gaps)[41,42], or adult stature (e.g., attaining a small size, so that only few forest strata can be occupied)[19]. We do acknowledge the effect of these processes on species abundances, although they are not directly analysed in this study.

In this study we use global datasets of tree composition of >22,000 forest plots and 11 traits and 2 multivariate trait axes of 1663 tree species (Fig. 1) to ask 1) how do locally dominant and rare tree species differ in their trait values in forest biomes across the globe?, and 2) how are these patterns driven by broadscale climatic gradients in temperature and water availability? We test the hypotheses that (i) dominant species express the more competitive trait values and locally dominant and rare tree species show a larger difference in traits in harsher environments, reflecting larger differences in habitat suitability and competitive ability between species[43], and that (ii) differences between trait values of locally dominant and rare tree species are more strongly driven by temperature than water availability because temperature influences a larger number of traits[25].

## Results

### Traits of dominant and rare tree species

The first PC axis, which included the traits of the locally dominant and rare tree species together, explained 41% of the variation and reflected a stem strategy spectrum ranging from angiosperms with wide vessels and thin bark to the left, to gymnosperms with narrow tracheids and thick bark to the right. The second PC axis explained 25% of the trait variation and was associated with traits related to photosynthetic carbon gain such as large crown diameter and high specific leaf area (SLA) (Fig. 2). Hence, differences amongst biomes, and more specifically the difference between angiosperms and gymnosperms, were particularly pronounced (Fig. 2).

Additionally, we compared traits of locally dominant and rare species per biome. Across biomes, dominant species had taller stems (Wilcoxon test, $39650337 > W > 475549$, $9721 > N > 1015$, $p < 0.01$), softer wood ($53798800 > W > 570861$, $p < 0.01$) and higher loadings on the first PC axis compared to rare species ($39237289 > W > 463234$,

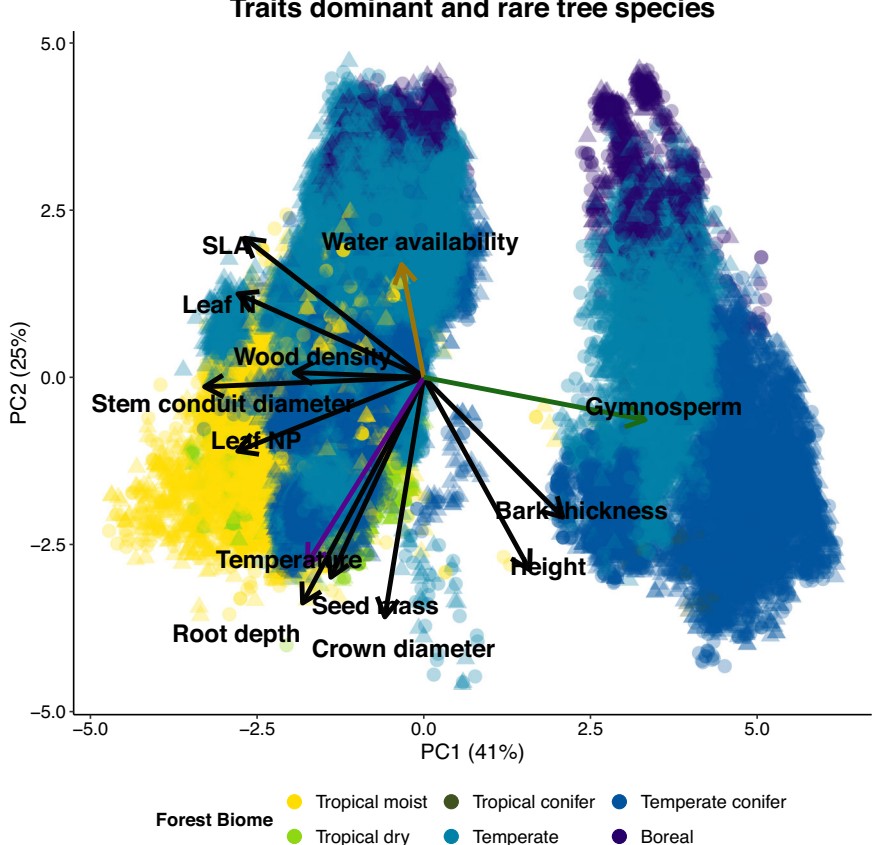

**Fig. 2 | A PCA visualizing 10 traits of dominant tree species (circles) and rare tree species (triangles).** The dominant and rare species per plot are visualised in this graph, and the circles and triangles represent therefore individual trees. The climatic variables temperature (purple arrow) and water availability (brown arrow), and gymnosperms (green arow) are as well indicated with an arrow. The six different forest biomes are visualized with different colours, see the legend for the colour explanation. The cluster on the left is dominated by angiosperms, while the cluster on the right is dominated by gymnosperms. For the same graph on species level, see Fig. S3A. Source data are provided as a Source Data file.

p < 0.01 (Fig. 3, Fig. S6 & Table S1). The other trait differences varied with biome, indicating that the environmental context selects for different traits affecting species abundance (Figs. 3 & S6). In temperate and boreal forests rare species had, compared to dominant species, deeper roots (49369516 > W > 1306032, 9721 > N > 1562, p < 0.001) and wider stem conduits (56055893 > W > 1163685, p < 0.001), while in the tropical biomes and temperate conifer forest rare species had a higher SLA (47709647 > W > 1579690, 8813 > N > 1015, p < 0.001) (Table S1). The absolute difference in trait values between locally dominant and rare species increases more than eightfold going from moist tropical forest (mean difference between scaled traits is 0.04 s.d.) to temperate conifer forest (mean difference between scaled traits is 0.3 s.d.), indicating that in harsher environments dominant and rare species differ more in their traits due to gymnosperm dominance (Fig. S6).

**Traits of dominant and rare tree species correlated with temperature and water availability**

Temperature showed the strongest correlation with rooting depth (Pearson correlation, r = 0.73, t = 330.95, N = 95659, p < 0.001), while water availability showed the strongest correlation with bark thickness (Pearson correlation, r = −0.38, t = −127.18, N = 95659, p < 0.001) (Fig. 2). We analysed the absolute difference of trait values between locally dominant and rare species along an environmental gradient of temperature and water availability. Temperature had a stronger effect on trait differences between dominant and rare species (mean variable importance is 34%) than water availability (mean variable importance is 11%) (Fisher's F-test, F = 2.19, N = 13, p < 0.001) (Figs. 4 & S7).

Interactions between temperature and water availability were often non-significant and had an average variable importance <3%. For all traits (except height), the difference between dominant and rare species showed hump-shaped or U-shaped relationships with temperature (Figs. 4 & S7), where the x-axis was crossed (indicating no differences between dominant and rare species) at a temperature between 5−8 °C, a maximum difference was attained around 15 °C, and trait difference became close to zero at high temperatures. For all traits (except height and PC1, PC2) the differences in trait values between dominant and rare species increases linearly with water availability and crosses the x axis at a water availability index of ca. 1.5 (Figs. 4 & S7). This coincides with the dominance of gymnosperms, which is higher at temperatures <4 °C and water availability >1.8.

## Discussion

In this study, we asked (1) how do locally dominant and rare tree species differ in their trait values in forest biomes across the globe?, and (2) how are these patterns driven by broad-scale climatic gradients in temperature and water availability? We found that in forests globally, dominant tree species grow taller, have softer wood and have higher loadings on the stem strategy axis (traits associated with gymnosperms) compared to rare species (Fig. 2). Locally dominant and rare species show a larger difference in traits in boreal compared to tropical forests (Fig. 3), and the differences in traits are more strongly driven by temperature than by water availability (Fig. 4).

Of the 11 traits and 2 multivariate trait axes evaluated, we found only three consistent trait differences across all biomes, and they are

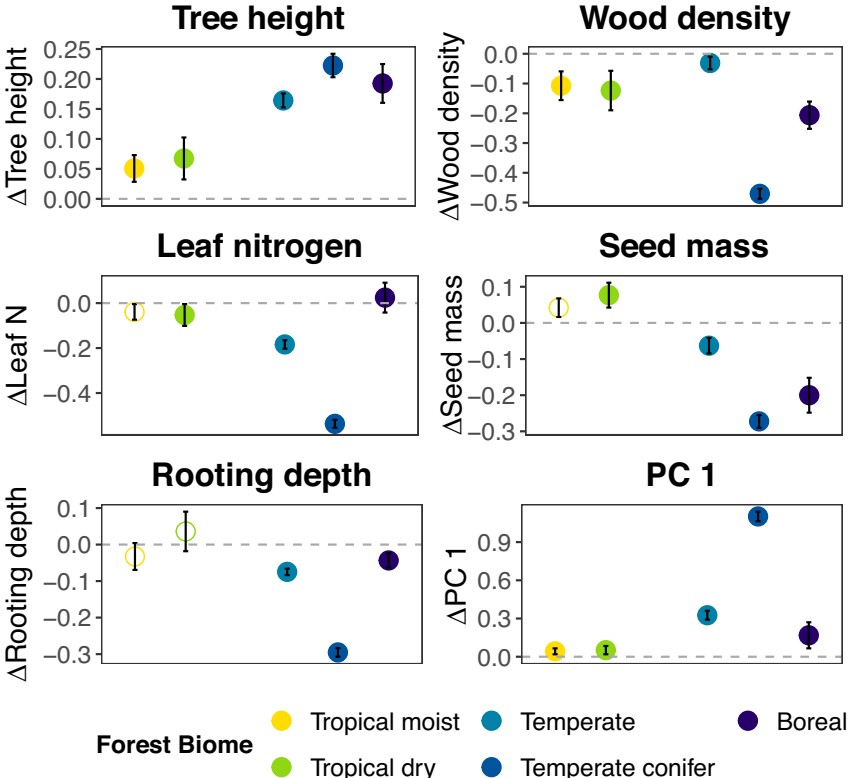

## Trait differences dominant and rare species

**Fig. 3 | The difference between standardized trait values of dominant and rare tree species per forest biome.** Ten traits, percentage gymnosperms and the first two PC axes are evaluated, of which five traits and PC1 are visualized here (see for the other traits and PC2 Fig. S6). The mean and confidence interval of the difference between trait values of dominant and rare species are displayed. If the mean is positive, dominant species have on average a higher trait value, while a negative mean indicates that rare species have a higher trait value. The grey dashed line indicates similar trait values between dominant and rare species. Closed dots indicate a significant difference between trait values of dominant and rare species (Wilcoxon test, $N > 1015$, $p < 0.05$). Detailed statistics per biome can be found in Table S1. Source data are provided as a Source Data file.

therefore globally important in determining local dominance and rarity. Dominant species are, compared to rare species, taller, have softer wood, and higher loadings on the multivariate stem strategy axis (soft wood, narrow conduit diameter), which are often characteristics of gymnosperm species (Figs. 2 & 3)[44,45]. Dominant species, therefore, seem to be canopy species and are fast and tall growers to secure light resources, leading to a taller stem height and lower wood density[19,46]. In contrast, rare species are likely understory or subcanopy species, and are shade-tolerant, slower-growing species with higher wood density[22]. The consistent trait differences between dominant and rare species might indicate that dominant species are earlier successional, faster growing species and that rare species are later successional slower growing species, a possible indication of human or natural disturbance in the forests evaluated in this study[47,48]. The difference in tree height is smallest for the tropical forest biomes and we found indeed that difference in height between dominant and rare tree species decreases with higher temperatures and water availability (Fig. 4). This might indicate that in the tropical biomes, dominant and rare species are present in multiple canopy strata, while in the temperate and boreal forests tend to have two canopy strata, where rare species are more restricted to the understory or subcanopy due to their small adult stature[49,50]. In sum, in forests globally, dominant tree species invest in faster growth and larger tree sizes compared to rare species.

For the other 8 traits and 1 multivariate trait axis, differences in trait values between locally dominant and rare species are dependent on the biome (Fig. 3 & S1), and therefore on the regional biotic and abiotic environmental context. These results are in line with studies across kingdoms of life, concluding that species abundances are related to certain trait values that depend on the environment[13,51–53]. Tropical dry and wet forests showed relatively few (4, 5) significant trait differences between dominant and rare species compared to other biomes, which is probably a reflection of the higher species diversity and, hence, functional redundancy in the tropics[54–56]. A higher functional redundancy may result in less striking trait differences between dominant and rare species, as species have more similar trait values[54]. Additionally, differences between dominant and rare species may be more difficult to detect because of the long tail of rare species in tropical forests, which may differ widely in their trait values[57]. Dominant tropical forest species had, next to the three traits mentioned above, also a lower SLA (Fig. S6), which may reflect stronger adaptations to shade for rare species, as moist tropical forests tend to be denser and continuously shaded compared to other forest biomes[58,59]. Regarding the dry tropical forests, a lower SLA could indicate a drought-tolerance strategy of the dominant, sun-exposed canopy trees[60–62].

Gymnosperms form a dominant component in temperate conifer and boreal forests, and are subordinates in temperate forests, which may explain why more traits (axes) differences in temperate and boreal forest biomes are significant (10–13) and larger compared to tropical biomes. The trait differences probably reflect a two-layered canopy structure, with a canopy layer occupied by dominants and an understory layer occupied by rare species (King et al.[50]). Gymnosperm trees have three unique features that set them apart from angiosperms

## Trait differences correlated with climate

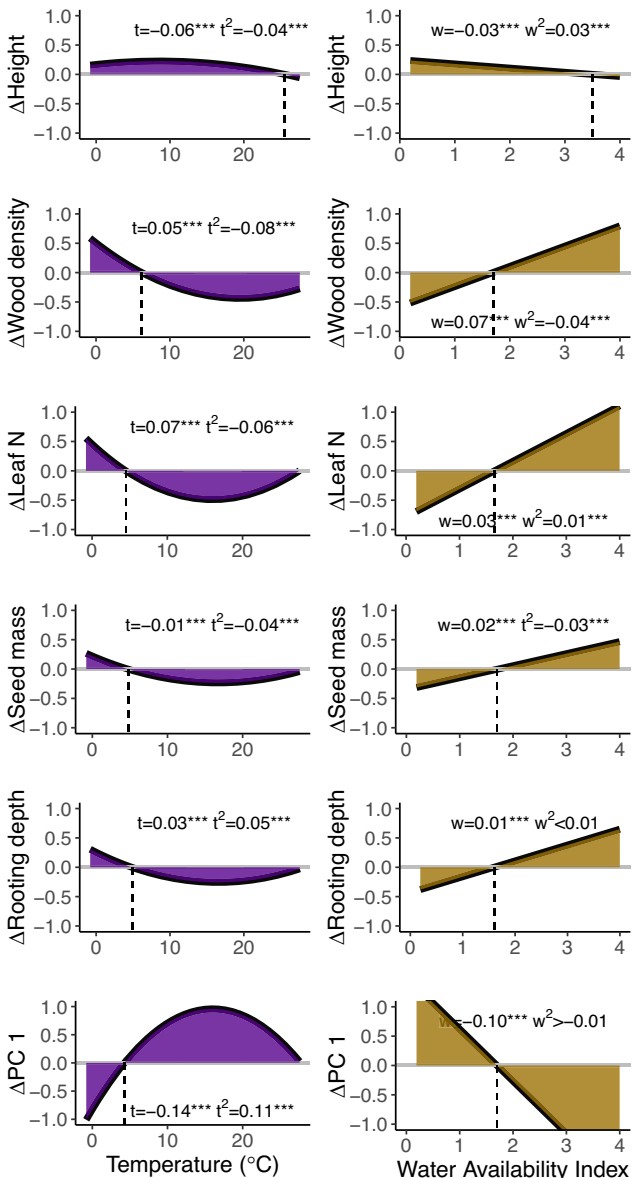

**Fig. 4 | The difference between five dominant and rare trait values and PC1 at the y-axis plotted against, respectively, mean annual temperature (left) and water availability (right).** See for the other traits and PC2 Fig. S7. Positive values indicate that trait values are higher for dominant species, while negative values indicate that trait values are higher for rare species. The graphs on temperature are modelled with a second-order polynomial function, the graphs on availability index are modelled with a linear function as this best fitted the data (See Supplementary Data 1). The standardized regression coefficients of the linear models are displayed of the climatic variables (t = temperature, w = water availability index) and squared climatic variables ($t^2$ = temperature², $w^2$ = water availability index²). The number of plots included in this analysis is 22,825. The significance of the regression coefficients is indicated with asterisks: *$p < 0.05$, **$p < 0.01$, ***$p < 0.001$. Source data is provided as a Source Data file.

and makes them well adapted to harsh environmental conditions such as low soil fertility, drought[63], pathogens[64,65] and mainly cold[44,45,66,67]. First, their water conducting conduits consist of narrow tracheids, which makes them more resistant against freezing- and drought-induced cavitation[65,68,69]. Second, mechanical strength is delivered by the relatively thick cell walls of their tracheids, and therefore they do not make (heavy) fibres, resulting in a low wood density[65]. Third, they

make structurally enforced needles, with low SLA and leaf nitrogen concentration that increases leaf longevity, and results in an evergreen leaf habit for nearly all gymnosperm species[70]. As a result, gymnosperm species differ strikingly from angiosperms in their traits and multivariate strategies (Fig. 2).

In general, trait differences are more driven by temperature compared to water availability, and the relationships between climate and trait differences seem to be mostly driven by adaptations to temperature-related productivity and biodiversity gradients, partly by angiosperm-gymnosperm differences, and to a lesser extent by drought adaptations. We found that temperature has a stronger effect on trait differences between locally dominant and rare species compared to water availability and that the interaction between temperature and water availability has a negligible effect (Figs. 4 & S7). These results indicate that it is mainly temperature that drives the range of trait values of dominant and rare species[25]. Temperature corresponds to the latitudinal gradients in species diversity, biomes, growing season length and productivity, while precipitation is less consistently related to latitude[71–74]. Therefore, in global analyses, temperature might influence a wider spectrum of traits, while water availability specifically has a stronger effect on traits related to drought resistance[25]. Indeed, we observed that water availability only has a stronger correlation with differences in bark thickness compared to temperature[75,76] (Fig. 4).

For all traits (except height), the difference between locally dominant and rare species showed hump-or U-shaped relationships with temperature (Figs. 4 & S7). At intermediate temperature conditions (around 15 °C, which coincides with temperate conifer forests) dominants and rare species show the most striking trait differences (Fig. S5). This probably reflects the trait differences between the dominant gymnosperm species and rare angiosperm species (Fig. S7). At colder temperatures (<5 °C, which coincides with boreal forests), the trait differences between dominant and rare species reverse, probably because this reflects a higher gymnosperm dominance (Fig. S7). At high temperatures (ca 25 °C, which coincides with tropical forests) the traits of dominant and rare species tend to converge, probably because of the high diversity and functional redundancy of tropical forests[77].

For all traits, the differences in trait values between locally dominant and rare species increased (nearly) linearly with water availability and crossed the x-axis at a water availability index of ca. 1.5 (Figs. 4 & S7). A higher water availability relates to an increase in productivity, forest height, density, complexity and species diversity, and hence, a stronger light competition[45,78–80]. In humid ecosystems, the dominant species show a more acquisitive strategy, as they are larger (larger crown diameter, rooting depth and higher seed mass)[46,81], have a higher water transport capacity (larger conduit diameter)[65,82], more productive leaves (larger SLA, leaf nitrogen concentration, and leaf N:P ratio)[70,83], and higher wood density (angiosperms)[65]. In arid systems, the dominant species are more conservative, as they are smaller, more drought resistant with narrow conduits or save water with a low SLA. These conservative trait values might indicate a survival strategy in arid systems, with traits adapted to drought to avoid cavitation[84].

Global databases of forest inventories are typically under sampled in more speciose tropical regions, potentially leading to an underestimation of trait variation among rare species at biome level. Nevertheless, we found that rare species in these regions (i.e., moist and dry tropical forests) already contain larger trait diversity than the dominant species, indicating that potential under sampling would not lead to a large bias of the overarching patterns (Fig. S8). The trait values used here were estimated based on phylogenetic and environmental information. This allowed for the incorporation of trait plasticity across environmental gradients, but it also introduces model-based uncertainty into the predictions. The imputation uncertainty has been shown to have negligible bias when averaging over many species[45]. Finally, there are processes shaping tree communities which

we did not consider in this study and are subject to future studies, such as metapopulation processes, historical (management) legacies, habitat specialization, and successional stage[19,36,38,42,85,86].

Ecosystems consist of dominant and rare species, which have their own unique contribution to ecosystem functioning. Here, we evaluated differences in trait values of locally dominant and rare tree species across global forests and explored how these trait profiles vary along broad environmental gradients. The difference between traits of dominant and rare species is more strongly driven by temperature compared to water availability, as temperature might affect a larger number of traits. Therefore, climate change driven global temperature rise may have a strong effect on the trait differences between dominant and rare tree species and may lead to strong community reassembly.

## Methods

### Forest inventory data

To identify dominant and rare species at plot level, we initially incorporated H1,2 million forest inventory plots sourced from the Global Forest Biodiversity Initiative (GFBI database). Each forest plot contains information on tree species richness, tree species abundance, year of measurement, plot size and location. Tree ferns and palms are not included in the database.

Plot sizes range from 0.0002 to 20 ha in the database and the plots include all trees with stem diameter at breast height ≥ 5 cm. As rare species are likely not captured accurately in very small plots, and trait variation is correlated with plot size as well[87], we excluded plot sizes smaller than the first quantile of 0.02 ha and outliers larger than 2 ha (in total 8.3% of the database). Additionally, plots measured before 1990 were filtered out, as these plots likely do not represent current forest composition and do not match with the climatic data we used (filtering out 21% of the remaining database). Also, trait values of the rare and dominant species change with successional forest age[85], we therefore excluded early successional plots with a forest age of less than 25 years, which corresponds to 1.3% of the remaining database[88]. Different forest age thresholds could potentially affect trait values of dominant and rare species. Yet, when comparing trait values using the 25-year threshold with 30- and 35-years thresholds we got very similar results (Fig. S11). Within the filtered database, the correlations between plot size and number of dominant species ($r = 0.23$, $r^2 = 0.05$, $p < 0.01$), and plot size and number of rare species was equally weakly related ($r = 0.23$, $r^2 = 0.05$, $p < 0.01$) (see for definition of dominant and rare species methods section *Identifying dominant and rare tree species*). See for an overview of the distribution of plot size within every forest biome Fig. S1 and the relationship between species abundances and plot size for this database[78,89]. In our filtered dataset, the mean plot size is 0.07 ha, the mean measurement year is 2004 and 53 years is the mean forest age, whereas 1.4% of the dataset is made up of old-growth forest (older than 140 years). Elimination of forest plots based on size, year of measurement, forest age, and incorporating only plots within the forest biomes[90], resulted in 660,552 plots in the filtered dataset (Fig. 1). Additionally, we incorporated only plots with six or more species, to clearly separate dominant and rare species, which included 23% of the filtered dataset based on plot size, measurement year and age.

Species names in the GFBI dataset were standardised using The Plant List[91]. 1.4% of the species names could not be matched using The Plant List, therefore subsequently the Global Biodiversity Information Facility (GBIF) backbone was sourced to standardize these species names to accepted species names[92]. In every forest plot the dominant and rare species were identified.

### Identifying dominant and rare tree species

Dominant and rare species were identified as the top and bottom 10% species according to the rank abundance curve at the plot level[93–96]. We defined dominant and rare species at the plot level, as this is the spatial scale at which species interact more directly with each other, and therefore the outcome of both abiotic and biotic interactions affecting species abundances is reflected reflected[97,98]. There are many different definitions of dominant and rare species, however, we choose the 10% most and least abundant individuals in terms of number of stems as this resembles the outer parts of the species abundance distribution. Also, the number of individuals is not automatically related to trait values, while for example, abundance based on basal area is related to the traits height, crown length and leaf area index[99–101]. As the definition of dominant and rare species could affect the results, a PCA was made where the dominant and rare species were classified as respectively the top and bottom 10% of the number of stems in a plot (Fig. S3B), showing a very similar pattern to Fig. 2. Additionally, dominant and rare species were defined as the top and bottom 5% and 15% of individuals in the plot, showing that the stricter the definition (e.g. lower percentage), the more plots are filtered out and the more pronounced the difference between dominant and rare species trait values are (Fig. S10). As described in Hordijk et al. (2024), both the dominant and rare species in the GFBI database are geographically widespread[89].

In forest plots containing between 6 and 19 species, the top two and bottom two species were defined as respectively dominant or rare, whereas for plots with >=20 species the 10% most and least abundant species were identified as dominant or rare, respectively. Additionally, if the rarest species in a plot comprised >10% of the sum of the stems, then this plot was excluded. This assured that the rare species are clearly distinguishable in abundance from the dominant species. Additionally, it also selected for the species-rich plots in the Boreal forest zone. Lastly, we randomly selected 10,000 plots from the temperate forest biome for the plot-level analyses, instead of a total of 135,043 plots, to reduce computation time and have a more balanced dataset representing the different forest biomes. The used subset of the temperate forest biome is a good representation of the traits of dominant and rare straits in this biome, which is verified with a bootstrapping procedure (Fig. S9).

### Trait selection

Using the trait imputation models of Maynard et al. (2022), a total of 18 trait values were computed for each tree occurrence in the GFBI dataset, encompassing a variety of leaf, wood and root traits with training data sourced from the TRY database[102]. In case the tree was identified up to genus level, the species-level average of that trait within that genus was calculated and used as an approximation of the trait value. These models incorporate intraspecific variation and thus provide a unique prediction of each trait for each of the 1663 species in each location where the species occurs, based on the combination of phylogenetic and environmental information. From every trait cluster identified by Maynard et al (2022), at least one trait was included for further analysis to guarantee sufficient statistical independence and to cover the range of plant life-history strategies. Ten traits were included reflecting the global Leaf-Height-Seed plant strategy scheme[103], and the global spectra of plant form and function[44]. The traits are related to the size of trees (tree height, crown diameter, rooting depth) and their propagules (seed dry mass), tissue construction costs (wood density, specific leaf area), hydraulics (stem conduit diameter), leaf economics and photosynthesis (leaf nitrogen per mass, leaf nitrogen/phosphorus ratio) and stem defence against disturbances such as fire and insects (bark thickness). For an overview of the traits and their ecological significance see Table 1. The trait values were log transformed (natural logarithm), to diminish the effect of outliers, and trait values were standardized to compare different trait units accurately in the analysis[104]. For the distribution of the untransformed trait values, see Fig. S2. Additionally, as gymnosperm and angiosperm species have distinct trait values (see also Fig. 2), the dominant and rare species were identified as either gymnosperm or angiosperm, based on their

**Table 1 | An overview of the ten traits and their ecological significance considered in this study**

| Trait | Ecological significance | References |
|---|---|---|
| Tree height | Taller trees intercept more light and their stature facilitates seed dispersal, which trade-offs against the increase in construction and maintenance costs and risk of breakage. | 110,111 |
| Rooting depth | Deeper roots enhance water uptake and tree stability, but also increases maintenance cost. | 112,113 |
| Specific leaf area (SLA) | A high SLA results in a short leaf lifespan and high carbon gain, and a low SLA is related to a long leaf lifespan and efficient nutrient conservation. | 16,23,24 |
| Stem conduit diameter | A wider stem conduit diameter results in more efficient water transportation, which increases plant productivity but also the risk of embolism. | 114,115 |
| Crown diameter | A larger crown diameter is related to a higher photosynthetic capacity, as more leaves are sun-exposed, but also increases the risk of branch damage. | 116 |
| Wood density | A higher wood density relates to better mechanical support, water transport and storage capacity of woody tissues, and is associated with slower growing species due to the energy investment. | 105 |
| Bark thickness | A thicker bark is related to water storage and fire protection, but results in a stiffer stem, which makes it more prone to stem breakage. | 117,118 |
| Leaf nitrogen | Leaf nitrogen reflects a trade-off between the benefits of a high photosynthetic potential with high nitrogen and the costs of acquiring nitrogen and suffering herbivory. | 119,120 |
| Leaf N/P ratio | A low N/P ratio indicate a high biomass production and a quick return on investments of carbon and nutrients, while a high N/P ratio promotes stress tolerance. | 70,121 |
| Seed mass | A higher seed mass results in a higher chance of seedling survival and at the same time a lower number of seeds produced and therefore a lower colonisation ability. | 122 |

family, and the percentage of dominant and rare species comprising gymnosperms was calculated per plot.

## Evaluating the difference between the traits of dominant and rare tree species

To evaluate general trade-offs between traits of the dominant and rare species, a PCA was performed including all dominant and rare species in the six different forest biomes; tropical moist forest, tropical dry forest, tropical conifer forest, temperate forest, temperate conifer forest and boreal forest[90] (Fig. 2). The loadings of the dominant and rare species per plot level on the first and second axes of the PCA were incorporated in further analyses. Among a total of 10 traits, the gymnosperm percentage and PC axes loadings, six are presented in the main text, to decrease the information displayed, and seven are presented in the supporting information. The results related to the traits tree height, wood density, leaf nitrogen concentration, seed mass, rooting depth, and the first PC axis are displayed in the main text as they represent a broad spectrum of traits related to different life history strategies[44,45,105]. To give an insight into the species-level differences, a PCA was made with dominant and rare species as data points in the PCA, rather than the different tree individuals (Fig. S3A). Additionally, to verify if the results are not only caused by chance or the way of calculating the dominant and rare trait values per plot, we randomised the data 100 times within the five main forest biomes, individually keeping the total number of trees per plot and the total number of individuals per tree species constant. Afterwards, we calculated the traits of the dominant and rare species in three different ways: using the trait mean, the median and the interquartile range. The null models indicate that, after randomization of the dataset, the three different ways of calculating trait values show a large to complete overlapping frequency distribution for each of the groups, indicating robust results using the median trait plot level value (Fig. S4).

For the dominant and rare species, the median trait value per plot was calculated, as otherwise, a different number of dominant or rare species could affect the difference in trait values. We calculated the difference in scaled trait value of the dominant and rare species by subtracting the scaled trait value of the rare species from the scaled trait value of the dominant species in the same plot. We included intraspecific variation in the analyses, since across plots trait values of the same species can fluctuate (see methods section *Trait selection*). With a Wilcoxon signed-rank test, the difference between the mean trait values was evaluated per forest biome. When evaluating the

differences between traits for dominant and rare species, the tropical conifer forest was excluded from the analyses due the low number of plots in this biome ($N = 182$) and therefore high standard error in trait differences between dominant and rare species.

## Evaluating the effect of temperature and water availability on trait differences

Climatic water availability was calculated as the ratio of mean annual precipitation over mean annual evapotranspiration at a resolution of 30 arc sec[106]. A climatic water availability <1 means a water-deficient arid environment, whereas a climatic water availability >1 corresponds to a more humid environment. Other studies also refer to this index as 'aridity index', but we prefer to use the term water availability, as a high value indicates a humid environment. Temperature is expressed as mean annual temperature, based on the CHELSA (Climatologies at high resolution for the earth's land surface areas) data at a resolution of 30 arc sec[107]. Across our dataset, temperature and water availability are weakly negatively correlated (Spearman correlation, $r = -0.09$, $r^2 = 0.008$, $N = 22{,}825$, $p < 0.001$), indicating that these two climatic variables represent distinct climatic gradients. An overview of the temperature and water availability range per biome can be found in Fig. S5.

The relationships between trait values and the two climatic variables were evaluated with a Pearson correlation. To explore the effects of temperature, water availability, and their interaction on differences in trait values, a second-order polynomial model was used, in which we corrected for plot size, forest age, elevation and biome. We evaluated the relationships with a second-order polynomial model as we expect that the relationships between trait differences and temperature or water availability can be a concave or convex relationship, as trait differences might be largest in the more extreme climatic conditions[43]. The independent variables in the model were scaled to a mean of zero and a standard deviation of one to facilitate comparability between the regression coefficients. To quantify the relative importance of temperature, water availability, their interaction, forest age, plot size, elevation and biome on trait differences, we used the scaled calc.relimp function in R[108]. This function evaluates the contribution of each independent variable to the variation explained by averaging the contribution of each independent variable to the $r^2$ in terms of its sum of squares across all possible fitting sequences. See Supplementary Data 1 for an overview of the regression coefficients and variable importance values. To evaluate the difference between the variable

importance of temperature and water availability, including both the non-transformed and squared variables, a two-sample t-test with equal variances was performed (Fisher's F-test, $N = 13$, $F = 2.19$, $p < 0.001$).

Data management and statistical analyses in this study were performed with the R-Studio interface to R[109].

## Reporting summary

Further information on research design is available in the Nature Portfolio Reporting Summary linked to this article.

## Data availability

The plot-level data of the difference between trait values of dominant and rare tree species are stored in Zenodo https://doi.org/10.5281/zenodo.15393651. The GFBI database is available upon written request at https://gfbinitiative.net/data/. Source data are provided with this paper.

## Code availability

The code used to perform the statistical analyses can be found in the Supplementary Code 1.

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

## Acknowledgements

This research has been funded by a grant from DOB Ecology. Swiss National Science Foundation, Ambizione grant #PZ00P3_193612 to DSM. JCS considers this work a contribution to Centre for Ecological Dynamics in a Novel Biosphere (ECONOVO), funded by Danish National Research Foundation (grant DNRF173), and his VILLUM Investigator project "Biodiversity Dynamics in a Changing World", funded by VILLUM FONDEN (grant 16549). The GFBI data from New Zealand were drawn from the Natural Forest plot data collected between January 2009 and March 2014 by the LUCAS programme for the New Zealand Ministry for the Environment and sourced from the New Zealand National Vegetation Survey Databank'. Russian Science Foundation Project 21-46-07002 for the plot data collected in the Krasnoyarsk region. Data from National Forest Inventory of Instituto de Conservação da Natureza (ICNF). FCT - Portuguese Foundation for Science and Technology, Project UIDB/ 04033/2020. GFBi plot data collection in the São Francisco de Paula National Forest, Rio Grande do Sul, Brazil was financed by Conselho Nacional de Desenvolvimento Científico e Tecnológico (CNPq)(project 520053/1998-2). ReVaTene project is funded by the Education and Research Ministry of Côte d'Ivoire, as part of the Debt Reduction-Development Contracts (C2Ds) managed by IRD GFBI data from

southern Ethiopia were collected with funding from the International Climate Initiative (IKI) of the German Federal Ministry for the Environment, Nature Conservation, Building and Nuclear Safety (BMU) (IKI-1 project number 09 II 066ETH A Kaffeewälder). GFBI data from Atlantic Forest, Brazil, was funded by the State of São Paulo Research Foundation (FAPESP 03/12595-7) as part of the BIOTA Programme. COTEC/IF 41.065/2005 and IBAMA/CGEN 093/2005 granted permits to establish the permanent plots and collect data. The Exploratory plots of FunDivEUROPE (with sites in Germany, Finland, Poland, Romania, Italy and Spain) received funding from the European Union Seventh Framework Programme (FP7/2007-2013) under grant agreement 265171. Permission to work in the MAWAS region of Indonesia: the BOS Foundation, the Indonesian Institute of Sciences (LIPI), the Direktorat Fasilitasi Organisasi Politik dan Kemasyarakatan, Departamen Dalam Negri, and the BKSDA Palangkaraya. Funding sources: The American Society of Primatologists, the Duke University Graduate School, the L.S.B. Leakey Foundation, the National Science Foundation (Grant no. 0452995), and the Wenner-Gren Foundation for Anthropological Research (Grant No. 7330). This study was supported by the National Natural Science Foundation of China (31800374), Shandong Provincial Natural Science Foundation (ZR2019BC083) The Spanish Agency for International Development Cooperation [Agencia Española de Cooperación Internacional para el Desarrollo (AECID)] and Fundación Biodiversidad, in cooperation with the governments of Syria and Lebanon. Projects D/9170/07, D/018222/08, D/023225/09 and D/032548/10 funded by the Spanish Agency for International Development Cooperation [Agencia Española de Cooperación Internacional para el Desarrollo (AECID)] and Fundación Biodiversidad, in cooperation with the Universidad Mayor de San Simón (UMSS), the FOMABO (Manejo Forestal en las Tierras Tropicales de Bolivia) project and CIMAL (Compañía Industrial Maderera Ltda.). All persons who made the Third Spanish Forest Inventory possible, especially the main coordinator, J. A. Villanueva (IFN3) The German Research Foundation (DFG) Priority Programme 1374 - Biodiversity Exploratories. Research was supported by APVV 20-0168 from the Slovak Research and Development Agency EC acknowledges funding from the project AdAgriF—Advanced methods of greenhouse gases emission reduction and sequestration in agriculture and forest landscape for climate change mitigation (CZ.02.01.01/00/22_008/0004635) We acknowledge collaboration with the International Boreal Forest Research Association (IBFRA, http://ibfra.org). We thank the Ministère des Forêts, de la Faune et des Parcs du Québec for access to their database of permanent sample plots. We thank the Amazon Forest Inventory Network (RAINFOR), the African Tropical Rainforest Observation Network, and the ForestPlots.net initiative for their contributions from Amazonian and African forests. These were supported by many projects, including an ERC Advanced Grant 291585 ("T-FORCES") and a Royal Society Wolfson Research Merit Award to O.L.P.; RAINFOR plots were additionally supported by the Gordon and Betty Moore Foundation and the UK Natural Environment Research Council (NERC), notably NERC Consortium Grants AMAZONICA (NE/F005806/1), TROBIT (NE/D005590/1), and BIO-RED (NE/N012542/1). This study was supported by GACR project 21-27454S from the Czech Science Foundation. Financial support from DBT, Govt. of India, through the project 'Mapping and quantitative assessment of geographic distribution and population status of plant resources of Eastern Himalayan region' is highly acknowledged. (Reference no. BT/PR7928/NDB/52/9/2006 dated 29.09.2006). Financial support from the Monafor network in Mexico was funded by many projects, including the National Forestry Commission (CONAFOR), Council of Science and Technology of the State of Durango (COCYTED), the Natural Environment Research Council, UK (NERC; NE/T011084/1), and local support of Ejidos and Comunidades. The French National Forest Inventory (NFI campaigns, raw data 2005 and following annual surveys) were downloaded by GFBI at https://inventaire-forestier.ign.fr/spip.php?rubrique159 (site accessed on 1 January 2015); the Italian Forest Inventory (2005 and 2015) were downloaded by GFBI at https://inventarioforestale.org/. GA was supported by Italian National Recovery Plan through the National Biodiversity Future Centre. Financial support from the Czech Science Foundation (Project no. 21-26883S). Plots in Mato Grosso, Brazil, were supported by the National Council for Scientific and Technological Development (CNPq), PELD-TRAN 441244/2016-5 and 441572/2020-0, and Mato Grosso State Research Support Foundation (FAPEMAT) – 0346321/2021. C.A.J. and S.V. acknowledge support from the Brazilian National Research Council/CNPq (PELD process 403710/2012–0); NERC and the State of São Paulo Research Foundation/FAPESP as part of the projects Functional Gradient, PELD/BIOTA and ECOFOR (processes 2003/12595-7, 2012/51509-8 and 2012/51872-5, within the BIOTA/FAPESP Programme (www.biota.org.br); COTEC/IF 002.766/2013 and 010.631/2013 permits. H.Y.H.C. acknowledges the support from NSERC (RGPIN-2019–05109 and STPGP428641) and the Canada Foundation for Innovation and Ontario Research Fund (CFI36014). K.S. acknowledges the support of data acqusition from project: "LIFE+ ForBioSensing PL Comprehensive monitoring of stand dynamics in Białowieża Forest supported with remote sensing techniques" which is co-funded by the EU Life Plus programme (contract number LIFE13 ENV/PL/000048) and The National Fund for Environmental Protection and Water Management in Poland (contract number 485/2014/WN10/OP-NM-LF/D) OB was supported by Romania National Council for Higher Education Funding, CNFIS, project number CNFIS-FDI-2024-F-0155. TMF was supported by a grant from the Czech Science Foundation (19-14620S). GFBi plot data collection in Santa Catarina, Brazil (FlorestaSC), was financed by Conselho Nacional de Desenvolvimento Científico e Tecnológico; FAPESC; SEMAE; FAO; SFB. RLC monitoring plots data in Costa Rica was supported by grants from the Andrew W. Mellon Foundation, the US National Science Foundation (NSF DEB-0424767, NSF DEB-0639393 and NSF DEB-1147429), US NASA Terrestrial Ecology Programme, and the University of Connecticut Research Foundation.

## Author contributions

Iris Hordijk, Tom W. Crowther and Daniel S. Maynard conceived of the study. Iris Hordijk extracted and analysed the data, and drafted the manuscript with assistance from Lourens Poorter, Daniel S. Maynard and Tom W. Crowther. Jingjing Liang, Peter B. Reich, Sergio de-Miguel, Gert-Jan Nabuurs, Javier G. P. Gamarra, Han Y. H. Chen, Mo Zhou, Susan K. Wiser, Hans Pretzsch, Alain Paquette, Nicolas Picard, Bruno Hérault, Jean-Francois Bastin, Giorgio Alberti, Meinrad Abegg, Yves C. Adou Yao, Angelica M. Almeyda Zambrano, Braulio V. Alvarado, Esteban Alvarez-Davila, Patricia Alvarez-Loayza, Luciana F. Alves, Iêda Amaral, Christian Ammer, Clara Antón-Fernández, Alejandro Araujo-Murakami, Luzmila Arroyo, Valerio Avitabile, Gerardo A. Aymard C., Timothy Baker, Olaf Banki, Jorcely Barroso, Meredith L. Bastian, Luca Birigazzi, Philippe Birnbaum, Robert Bitariho, Pascal Boeckx, Frans Bongers, Olivier Bouriaud, Pedro H. S. Brancalion, Susanne Brandl, Francis Q. Brearley, Roel Brienen, Eben N. Broadbent, Helge Bruelheide, Roberto Cazzolla Gatti, Ricardo G. Cesar, Goran Cesljar, Robin L. Chazdon, Chelsea Chisholm, Emil Cienciala, Connie J. Clark, David B. Clark, Gabriel Colletta, David Coomes, Fernando Cornejo Valverde, Jose J. Corral-Rivas, Philip Crim, Jonathan Cumming, Selvadurai Dayanandan, André L. de Gasper, Mathieu Decuyper, Géraldine Derroire, Ben DeVries, Ilija Djordjevic, Aurélie Dourdain, Jiri Dolezal, Nestor Laurier Engone Obiang, Brian Enquist, Teresa Eyre, Adandé Belarmain Fandohan, Tom M. Fayle, Leandro V. Ferreira, Ted R. Feldpausch, Leena Finér, Markus Fischer, Christine Fletcher, Lorenzo Frizzera, Damiano Gianelle, Henry B. Glick, David Harris, Andrew Hector, Andreas Hemp, John Herbohn, Annika Hillers, Eurídice N. Honorio Coronado, Cang Hui, Hyunkook Cho, Thomas Ibanez, Ilbin Jung, Nobuo Imai, Andrzej M. Jagodzinski, Bogdan Jaroszewicz, Vivian Johannsen, Carlos A. Joly, Tommaso Jucker, Viktor Karminov, Kuswata Kartawinata, Elizabeth Kearsley, David Kenfack, Deborah Kennard, Sebastian Kepfer-Rojas, Gunnar Keppel, Mohammed Latif Khan, Timothy Killeen, Hyun Seok Kim, Kanehiro Kitayama, Michael

# Article

Köhl, Henn Korjus, Florian Kraxner, Diana Laarmann, Mait Lang, Simon Lewis, Huicui Lu, Natalia Lukina, Brian Maitner, Yadvinder Malhi, Eric Marcon, Beatriz Schwantes Marimon, Ben Hur Marimon-Junior, Andrew Robert Marshall, Emanuel Martin, Olga Martynenko, Jorge A. Meave, Omar Melo-Cruz, Casimiro Mendoza, Cory Merow, Stanislaw Miscicki, Abel Monteagudo Mendoza, Vanessa Moreno, Sharif A. Mukul, Philip Mundhenk, Maria G. Nava-Miranda, David Neill, Victor Neldner, Radovan Nevenic, Michael Ngugi, Pascal A. Niklaus, Jacek Oleksyn, Petr Ontikov, Edgar Ortiz-Malavasi, Yude Pan, Alexander Parada-Gutierrez, Elena Parfenova, Minjee Park, Marc Parren, Narayanaswamy Parthasarathy, Pablo L. Peri, Sebastian Pfautsch, Oliver L. Phillips, Maria Teresa Piedade, Daniel Piotto, Nigel C. A. Pitman, Martina Pollastrini, Irina Polo, Axel Dalberg Poulsen, John R. Poulsen, Freddy Ramirez Arevalo, Zorayda Restrepo-Correa, Mirco Rodeghiero, Samir Rolim, Anand Roopsind, Francesco Rovero, Ervan Rutishauser, Purabi Saikia, Christian Salas-Eljatib, Peter Schall, Dmitry Schepaschenko, Michael Scherer-Lorenzen, Bernhard Schmid, Jochen Schöngart, Eric B. Searle, Vladimír Seben, Federico Selvi, Josep M. Serra-Diaz, Douglas Sheil, Anatoly Shvidenko, Javier Silva-Espejo, Marcos Silveira, James Singh, Plinio Sist, Ferry Slik, Bonaventure Sonké, Alexandre F. Souza, Hans ter Steege, Krzysztof Stereńczak, Jens-Christian Svenning, Miroslav Svoboda, Ben Swanepoel, Natalia Targhetta, Nadja Tchebakova, Raquel Thomas, Elena Tikhonova, Peter Umunay, Vladimir Usoltsev, Renato Valencia, Fernando Valladares, Fons van der Plas, Tran Van Do, Michael E. Van Nuland, Rodolfo Vasquez Martinez, Hans Verbeeck, Helder Viana, Alexander C. Vibrans, Simone Vieira, Klaus von Gadow, Hua-Feng Wang, James Watson, Gijsbert D. A. Werner, Florian Wittmann, Verginia Wortel, Roderick Zagt, Tomasz Zawila-Niedzwiecki, Chunyu Zhang, Xiuhai Zhao, Zhi-Xin Zhu and Irie Casimir Zo-Bi provided data for the analysis. All authors assisted with revisions and gave final approval for publication.

## Competing interests

The authors declare no competing interests.

## Additional information

Iris Hordijk [1,2] ✉, Lourens Poorter [2], Jingjing Liang [3], Peter B. Reich [4,5], Sergio de-Miguel [6,7], Gert-Jan Nabuurs [2], Javier G. P. Gamarra [8], Han Y. H. Chen [9], Mo Zhou[3], Susan K. Wiser [10], Hans Pretzsch[11], Alain Paquette [12], Nicolas Picard[13], Bruno Hérault [14,15], Jean-Francois Bastin [16], Giorgio Alberti [17,18], Meinrad Abegg [19], Yves C. Adou Yao [20], Angelica M. Almeyda Zambrano[21], Braulio V. Alvarado[22], Esteban Alvarez-Davila[23], Patricia Alvarez-Loayza[24], Luciana F. Alves [25], Iêda Amaral[26], Christian Ammer [27], Clara Antón-Fernández [28], Alejandro Araujo-Murakami[29], Luzmila Arroyo[29], Valerio Avitabile[30], Gerardo A. Aymard C [31,32], Timothy Baker [33], Olaf Banki[34], Jorcely Barroso[35], Meredith L. Bastian [36,37], Luca Birigazzi[38], Philippe Birnbaum [39], Robert Bitariho[40], Pascal Boeckx [41], Frans Bongers [2], Olivier Bouriaud [42], Pedro H. S. Brancalion [43], Susanne Brandl[44], Francis Q. Brearley[45], Roel Brienen [33], Eben N. Broadbent [46], Helge Bruelheide [47,48], Roberto Cazzolla Gatti[49], Ricardo G. Cesar[43], Goran Cesljar[50], Robin L. Chazdon [51,52], Chelsea Chisholm[1], Emil Cienciala [53,54], Connie J. Clark[55], David B. Clark[56], Gabriel Colletta[57], David Coomes [58], Fernando Cornejo Valverde[59], Jose J. Corral-Rivas [60], Philip Crim[61,62], Jonathan Cumming[62], Selvadurai Dayanandan[63], André L. de Gasper [64], Mathieu Decuyper [2], Géraldine Derroire [65], Ben DeVries [66], Ilija Djordjevic[67], Aurélie Dourdain[65], Jiri Dolezal [68,69], Nestor Laurier Engone Obiang[70], Brian Enquist [71,72], Teresa Eyre[73], Adandé Belarmain Fandohan[74], Tom M. Fayle[75,76,77], Leandro V. Ferreira[78], Ted R. Feldpausch [79], Leena Finér[80], Markus Fischer [81], Christine Fletcher[82], Lorenzo Frizzera[83], Damiano Gianelle [83], Henry B. Glick [84], David Harris [85], Andrew Hector[86], Andreas Hemp [87], John Herbohn[52], Annika Hillers[88,89], Eurídice N. Honorio Coronado [90], Cang Hui [91,92], Hyunkook Cho[93], Thomas Ibanez[39], Ilbin Jung[93], Nobuo Imai[94], Andrzej M. Jagodzinski [95,96], Bogdan Jaroszewicz [97], Vivian Johannsen [98], Carlos A. Joly[99], Tommaso Jucker [100], Viktor Karminov [101], Kuswata Kartawinata[24], Elizabeth Kearsley[102], David Kenfack[103], Deborah Kennard[104], Sebastian Kepfer-Rojas [98], Gunnar Keppel [105], Mohammed Latif Khan [106], Timothy Killeen[29], Hyun Seok Kim [107,108,109,110], Kanehiro Kitayama[111], Michael Köhl [112], Henn Korjus [113], Florian Kraxner [114], Diana Laarmann[113], Mait Lang[113], Simon Lewis [33,115], Huicui Lu[116], Natalia Lukina [117], Brian Maitner [72], Yadvinder Malhi[118], Eric Marcon [119], Beatriz Schwantes Marimon[120], Ben Hur Marimon-Junior[120],

Andrew Robert Marshall[52,121,122], Emanuel Martin[123], Olga Martynenko[124], Jorge A. Meave [125], Omar Melo-Cruz[126], Casimiro Mendoza[127], Cory Merow [51], Stanislaw Miscicki[128], Abel Monteagudo Mendoza[129,130], Vanessa Moreno[43], Sharif A. Mukul [52,131], Philip Mundhenk[112], Maria G. Nava-Miranda [132,133], David Neill [134], Victor Neldner [73], Radovan Nevenic[67], Michael Ngugi[73], Pascal A. Niklaus [135], Jacek Oleksyn[95], Petr Ontikov[101], Edgar Ortiz-Malavasi[22], Yude Pan [136], Alexander Parada-Gutierrez[29], Elena Parfenova [137], Minjee Park[3,107], Marc Parren[138], Narayanaswamy Parthasarathy[139], Pablo L. Peri [140], Sebastian Pfautsch [141], Oliver L. Phillips [33], Maria Teresa Piedade [142], Daniel Piotto [143], Nigel C. A. Pitman [24], Martina Pollastrini [144], Irina Polo[145], Axel Dalberg Poulsen [85], John R. Poulsen [55], Freddy Ramirez Arevalo[146], Zorayda Restrepo-Correa[147], Mirco Rodeghiero[148], Samir Rolim [143], Anand Roopsind[149], Francesco Rovero [150,151], Ervan Rutishauser[152], Purabi Saikia [153], Christian Salas-Eljatib [154,155,156], Peter Schall [27], Dmitry Schepaschenko [114], Michael Scherer-Lorenzen [157], Bernhard Schmid [135], Jochen Schöngart[142], Eric B. Searle [12], Vladimír Seben [158], Federico Selvi [144], Josep M. Serra-Diaz [159,160], Douglas Sheil [2,161], Anatoly Shvidenko [114], Javier Silva-Espejo[162], Marcos Silveira [163], James Singh[164], Plinio Sist [14], Ferry Slik[165], Bonaventure Sonké[166], Alexandre F. Souza[167], Hans ter Steege [34,168], Krzysztof Stereńczak [169], Jens-Christian Svenning [160,170], Miroslav Svoboda [171], Ben Swanepoel[172], Natalia Targhetta[142], Nadja Tchebakova [137], Raquel Thomas[173], Elena Tikhonova [118], Peter Umunay [84], Vladimir Usoltsev[174], Renato Valencia[175], Fernando Valladares[176], Fons van der Plas[177], Tran Van Do [178], Michael E. Van Nuland [179], Rodolfo Vasquez Martinez [128], Hans Verbeeck [102], Helder Viana [180,181], Alexander C. Vibrans [54,182], Simone Vieira [183], Klaus von Gadow [184], Hua-Feng Wang[185], James Watson[186], Gijsbert D. A. Werner[187], Florian Wittmann [188], Verginia Wortel [189], Roderick Zagt[190], Tomasz Zawila-Niedzwiecki[191], Chunyu Zhang [192], Xiuhai Zhao [192], Zhi-Xin Zhu[185], Irie Casimir Zo-Bi[193], Daniel S. Maynard [1,194] & Thomas W. Crowther [1]

[1]Institute of Integrative Biology, ETH Zurich (Swiss Federal Institute of Technology), Zurich, Switzerland. [2]Wageningen University and Research, Wageningen, The Netherlands. [3]Department of Forestry and Natural Resources, Purdue University, West Lafayette, IN, USA. [4]Department of Forest Resources, University of Minnesota, St Paul, MN, USA. [5]Hawkesbury Institute for the Environment, Western Sydney University, Penrith, NSW, Australia. [6]Department of Agricultural and Forest Sciences and Engineering, University of Lleida, Lleida, Spain. [7]Forest Science and Technology Centre of Catalonia (CTFC), Solsona, Spain. [8]Forestry Division, Food and Agriculture Organization of the United Nations, Rome, Italy. [9]Faculty of Natural Resources Management, Lakehead University, Thunder Bay, ON, Canada. [10]Manaaki Whenua–Landcare Research, Lincoln, New Zealand. [11]Chair for Forest Growth and Yield Science, TUM School for Life Sciences, Technical University of Munich, Munich, Germany. [12]Centre for Forest Research, Université du Québec à Montréal, Montréal, QC, Canada. [13]GIP ECOFOR, Paris, France. [14]CIRAD, Forêts et Sociétés, Montpellier, France. [15]Forêts et Sociétés, Univ Montpellier, CIRAD, Montpellier, France. [16]Gembloux Agro Bio-Tech, University of Liege, Liege, Belgium. [17]Faculty of Science and Technology, Free University of Bolzano, Bolzano, Italy. [18]Department of Agricultural, Food, Environmental and Animal Sciences, University of Udine, Udine, Italy. [19]Swiss Federal Institute for Forest, Snow and Landscape Research, WSL, Birmensdorf, Switzerland. [20]UFR Biosciences, University Félix Houphouët-Boigny, Abidjan, Côte d'Ivoire. [21]Spatial Ecology and Conservation Laboratory, Center for Latin American Studies, University of Florida, Gainesville, FL, USA. [22]Forestry School, Tecnológico de Costa Rica TEC, Cartago, Costa Rica. [23]Fundacion ConVida, Universidad Nacional Abierta y a Distancia, UNAD, Medellin, Colombia. [24]Field Museum of Natural History, Chicago, IL, USA. [25]Center for Tropical Research, Institute of the Environment and Sustainability, UCLA, Los Angeles, CA, USA. [26]National Institute of Amazonian Research, Manaus, Brazil. [27]Silviculture and Forest Ecology of the Temperate Zones, University of Göttingen, Göttingen, Germany. [28]Division of Forest and Forest Resources, Norwegian Institute of Bioeconomy Research (NIBIO), Ås, Norway. [29]Museo de Historia natural Noel kempff Mercado, Santa Cruz, Bolivia. [30]European Commission, Joint Research Centre, Ispra, Italy. [31]UNELLEZ-Guanare, Programa de Ciencias del Agro y el Mar, Herbario Universitario (PORT), Portuguesa, Venezuela. [32]Compensation International S. A. Ci Progress-GreenLife, Bogotá, D.C., Colombia. [33]School of Geography, University of Leeds, Leeds, UK. [34]Naturalis Biodiversity Centre, Leiden, The Netherlands. [35]Centro Multidisciplinar, Universidade Federal do Acre, Rio Branco, Brazil. [36]Proceedings of the National Academy of Sciences, Washington, DC, USA. [37]Department of Evolutionary Anthropology, Duke University, Durham, NC, USA. [38]United Nation Framework Convention on Climate Change, Bonn, Germany. [39]AMAP, Univ Montpellier, CIRAD, CNRS, INRAE, IRD, Montpellier, France. [40]Institute of Tropical Forest Conservation, Mbarara University of Sciences and Technology, Mbarara, Uganda. [41]Isotope Bioscience Laboratory - ISOFYS, Ghent University, Ghent, Belgium. [42]Stefan cel Mare University of Suceava, Suceava, Romania. [43]Department of Forest Sciences, Luiz de Queiroz College of Agriculture, University of São Paulo, Piracicaba, Brazil. [44]Bavarian State Institute of Forestry, Freising, Germany. [45]Department of Natural Sciences, Manchester Metro-politan University, Manchester, UK. [46]Spatial Ecology and Conservation Laboratory, School of Forest, Fisheries, and Geomatics Sciences, University of Florida, Gainesville, FL, USA. [47]Institute of Biology, Geobotany and Botanical Garden, Martin Luther University Halle-Wittenberg, Halle-, Wittenberg, Germany. [48]German Centre for Integrative Biodiversity Research (iDiv) Halle-Jena-Leipzig, Leipzig, Germany. [49]Biological Institute, Tomsk State University, Tomsk, Russia. [50]Department of Spatial Regulation, GIS and Forest Policy, Institute of Forestry, Belgrade, Serbia. [51]Department of Ecology and Evolutionary Biology, University of Connecticut, Storrs, CT, USA. [52]Tropical Forests and People Research Centre, University of the Sunshine Coast, Maroochydore, QL, Australia. [53]IFER - Institute of Forest Ecosystem Research, Jilove u Prahy, Czech Republic. [54]Global Change Research Institute CAS, Brno, Czech Republic. [55]Nicholas School of the Environment, Duke University, Durham, NC, USA. [56]Department of Biology, University of Missouri-St Louis, St Louis, MO, USA. [57]Programa de Pós-graduação em Biologia Vegetal, Instituto de Biologia, Universidade Estadual de Campinas, Campinas, Brazil. [58]Department of Plant Sciences and Conservation Research Institute, University of Cambridge, Cambridge, UK. [59]Andes to Amazon Biodiversity Program, Madre de Dios, Peru. [60]Facultad de Ciencias Forestales, Universidad Juárez del Estado de Durango, Durango, Mexico. [61]Department of Physical and Biological Sciences, The College of Saint Rose, Albany, NY, USA. [62]Department of Biology, West Virginia University, Morgantown, WV, USA. [63]Biology Department, Centre for Structural and Functional Genomics, Concordia University, Montreal, QC, Canada. [64]Natural Science Department, Universidade Regional de Blumenau, Blumenau, Brazil. [65]Cirad, UMR EcoFoG (AgroParistech, CNRS, INRAE, Université des Antilles, Université de la Guyane), Kourou, French Guiana. [66]Department of Geographical Sciences, University of Maryland, College Park, MD, USA. [67]Institute of Forestry, Belgrade, Serbia. [68]Institute of Botany, The Czech Academy

of Sciences, 25243 Průhonice, Czech Republic. ⁶⁹Department of Botany, Faculty of Science, University of South Bohemia, České Budějovice, Czech Republic. ⁷⁰IRET, Herbier National du Gabon (CENAREST), Libreville, Gabon. ⁷¹Department of Ecology and Evolutionary Biology, University of Arizona, Tucson, AZ, USA. ⁷²The Santa Fe Institute, Santa Fe, NM, USA. ⁷³Queensland Herbarium, Department of Environment and Science, Toowong, QL, Australia. ⁷⁴Ecole de Foresterie et Ingénierie du Bois, Université Nationale d'Agriculture, Ketou, Benin. ⁷⁵School of Biological and Behavioural Sciences, Queen Mary University of London, London, UK. ⁷⁶Biology Centre of the Czech Academy of Sciences, Institute of Entomology, Ceske Budejovice, Czech Republic. ⁷⁷Institute for Tropical Biology and Conservation, Universiti Malaysia Sabah, Kota Kinabalu, Sabah, Malaysia. ⁷⁸Museu Paraense Emílio Goeldi. Coordenação de Ciências da Terra e Ecologia, Belém, Pará, Brasil. ⁷⁹Geography, College of Life and Environmental Sciences, University of Exeter, Exeter, UK. ⁸⁰Natural Resources Institute Finland (Luke), Joensuu, Finland. ⁸¹Institute of Plant Sciences, University of Bern, Bern, Switzerland. ⁸²Forest Research Institute Malaysia, Kuala Lumpur, Malaysia. ⁸³Research and Innovation Center, Fondazione Edmund Mach, San Michele all'Adige, Italy. ⁸⁴School of Forestry and Environmental Studies, Yale University, New Haven, CT, USA. ⁸⁵Royal Botanic Garden Edinburgh, Edinburgh, UK. ⁸⁶Department of Plant Sciences, University of Oxford, Oxford, UK. ⁸⁷Department of Plant Systematics, University of Bayreuth, Bayreuth, Germany. ⁸⁸Centre for Conservation Science, The Royal Society for the Protection of Birds, Sandy, UK. ⁸⁹Wild Chimpanzee Foundation, Liberia Office, Monrovia, Liberia. ⁹⁰School of Geography and Sustainable Development, University of St Andrews, St Andrews, UK. ⁹¹Centre for Invasion Biology, Department of Mathematical Sciences, Stellenbosch University, Stellenbosch, South Africa. ⁹²Theoretical Ecology Unit, African Institute for Mathematical Sciences, Cape Town, South Africa. ⁹³Division of Forest Resources Information, Korea Forest Promotion Institute, Seoul, South Korea. ⁹⁴Department of Forest Science, Tokyo University of Agriculture, Tokyo, Japan. ⁹⁵Institute of Dendrology, Polish Academy of Sciences, Kórnik, Poland. ⁹⁶Poznań University of Life Sciences, Department of Game Management and Forest Protection, Poznań, Poland. ⁹⁷Faculty of Biology, Białowieża Geobotanical Station, University of Warsaw, Białowieża, Poland. ⁹⁸Department of Geosciences and Natural Resource Management, University of Copenhagen, Copenhagen, Denmark. ⁹⁹Department of Plant Biology, Institute of Biology, University of Campinas, UNICAMP, Campinas, Brazil. ¹⁰⁰School of Biological Sciences, University of Bristol, Bristol, UK. ¹⁰¹Bauman Moscow State Technical University, Mytischi, Russian Federation. ¹⁰²CAVElab-Computational and Applied Vegetation Ecology, Department of Environment, Ghent University, Ghent, Belgium. ¹⁰³CTFS-ForestGEO, Smithsonian Tropical Research Institute, Balboa, Panama. ¹⁰⁴Department of Physical and Environmental Sciences, Colorado Mesa University, Grand Junction, CO, USA. ¹⁰⁵UniSA STEM and Future Industries Institute, University of South Australia, Adelaide, SA, Australia. ¹⁰⁶Department of Botany, Dr Harisingh Gour Vishwavidyalaya (A Central University), Sagar, MP, India. ¹⁰⁷Department of Agriculture, Forestry and Bioresources, Seoul National University, Seoul, South Korea. ¹⁰⁸Interdisciplinary Program in Agricultural and Forest Meteorology, Seoul National University, Seoul, South Korea. ¹⁰⁹National Center for Agro Meteorology, Seoul, South Korea. ¹¹⁰Research Institute for Agriculture and Life Sciences, Seoul National University, Seoul, South Korea. ¹¹¹Graduate School of Agriculture, Kyoto University, Kyoto, Japan. ¹¹²Institute for World Forestry, University of Hamburg, Hamburg, Germany. ¹¹³Institute of Forestry and Rural Engineering, Estonian University of Life Sciences, Tartu, Estonia. ¹¹⁴International Institute for Applied Systems Analysis, Laxenburg, Austria. ¹¹⁵Department of Geography, University College London, London, UK. ¹¹⁶Faculty of Forestry, Qingdao Agricultural University, Qingdao, China. ¹¹⁷Center for Forest Ecology and Productivity, Russian Academy of Sciences, Moscow, Russia. ¹¹⁸School of Geography, University of Oxford, Oxford, UK. ¹¹⁹UMR EcoFoG, AgroParisTech, Kourou, France. ¹²⁰Programa de Pós-graduação em Ecologia e Conservação, Universidade do Estado de Mato Grosso, Nova Xavantina, Brazil. ¹²¹Flamingo Land Ltd, Kirby Misperton, UK. ¹²²Department of Environment & Geography, University of York, York, UK. ¹²³Department of Wildlife Management, College of African Wildlife Management, Mweka, Tanzania. ¹²⁴All-Russian Institute of Continuous Education in Forestry, Pushkino, Russian Federation. ¹²⁵Departamento de Ecología y Recursos Naturales, Facultad de Ciencias, Universidad Nacional Autónoma de México, Mexico City, Mexico. ¹²⁶Universidad del Tolima, Ibagué, Colombia. ¹²⁷Colegio de Profesionales Forestales de Cochabamba, Cochabamba, Bolivia. ¹²⁸Warsaw University of Life Sciences, Department of Forest Management, Dendrometry and Forest Economics, Warsaw, Poland. ¹²⁹Jardín Botánico de Missouri, Oxapampa, Peru. ¹³⁰Universidad Nacional de San Antonio Abad del Cusco, Cusco, Peru. ¹³¹Department of Environment and Development Studies, United International University, Dhaka, Bangladesh. ¹³²Colegio de Ciencias y Humanidades. Universidad Juárez del Estado de Durango, Durango, Mexico. ¹³³Escuela Politécnica Superior de Ingeniería. Campus Terra. Universidad de Santiago de Compostela, Lugo, Spain. ¹³⁴Universidad Estatal Amazónica, Puyo, Pastaza, Ecuador. ¹³⁵Department of Evolutionary Biology and Environmental Studies, University of Zürich, Zürich, Switzerland. ¹³⁶Climate, Fire, and Carbon Cycle Sciences, USDA Forest Service, Durham, NC, USA. ¹³⁷V. N. Sukachev Institute of Forest, FRC KSC, Siberian Branch of the Russian Academy of Sciences, Krasnoyarsk, Russia. ¹³⁸Forest Ecology and Forest Management Group, Wageningen University & Research, Wageningen, The Netherlands. ¹³⁹Department of Ecology and Environmental Sciences, Pondicherry University, Puducherry, India. ¹⁴⁰Instituto Nacional de Tecnología Agropecuaria (INTA), Universidad Nacional de la Patagonia Austral (UNPA), Consejo Nacional de Investigaciones Científicas y Tecnicas (CONICET), Rio Gallegos, Argentina. ¹⁴¹School of Social Sciences (Urban Studies), Western Sydney University, Penrith, NSW, Australia. ¹⁴²Instituto Nacional de Pesquisas da Amazônia, Manaus, Brazil. ¹⁴³Laboratório de Dendrologia e Silvicultura Tropical, Centro de Formação em Ciências Agroflorestais, Universidade Federal do Sul da Bahia, Itabuna, Brazil. ¹⁴⁴Department of Agriculture, Food, Environment and Forest (DAGRI), University of Firenze, Florence, Italy. ¹⁴⁵Jardín Botánico de Medellín, Medellín, Colombia. ¹⁴⁶Universidad Nacional de la Amazonía Peruana, Iquitos, Peru. ¹⁴⁷Servicios Ecosistémicos y Cambio Climático (SECC), Fundación Con Vida & Corporación COL-TREE, Medellín, Colombia. ¹⁴⁸Centro Agricoltura, Alimenti, Ambiente, University of Trento, San Michele all'Adige, Italy. ¹⁴⁹Department of Biological Sciences, Boise State University, Boise, ID, USA. ¹⁵⁰Department of Biology, University of Florence, Florence, Italy. ¹⁵¹Tropical Biodiversity, MUSE - Museo delle Scienze, Trento, Italy. ¹⁵²Info Flora, Geneva, Switzerland. ¹⁵³Department of Environmental Sciences, Central University of Jharkhand, Ranchi, India. ¹⁵⁴Centro de Modelación y Monitoreo de Ecosistemas, Universidad Mayor, Santiago, Chile. ¹⁵⁵Vicerrectoría de Investigación y Postgrado, Universidad de La Frontera, Temuco, Chile. ¹⁵⁶Departamento de Silvicultura y Conservación de la Naturaleza, Universidad de Chile, Santiago, Chile. ¹⁵⁷Faculty of Biology, Geobotany, University of Freiburg, Freiburg im Breisgau, Germany. ¹⁵⁸National Forest Centre, Forest Research Institute Zvolen, Zvolen, Slovakia. ¹⁵⁹Université de Lorraine, AgroParisTech, Inra, Silva, Nancy, France. ¹⁶⁰Center for Ecological Dynamics in a Novel Biosphere (ECONOVO) & Center for Biodiversity Dynamics in a Changing World (BIOCHANGE), Department of Biology, Aarhus University, Ny Munkegade, Denmark. ¹⁶¹Faculty of Environmental Sciences and Natural Resource Management, Norwegian University of Life Sciences, Ås, Norway. ¹⁶²Departamento de Biología, Universidad de la Serena, La Serena, Chile. ¹⁶³Centro de Ciências Biológicas e da Natureza, Universidade Federal do Acre, Rio Branco, Acre, Brazil. ¹⁶⁴Guyana Forestry Commission, Georgetown, French Guiana. ¹⁶⁵Environmental and Life Sciences, Faculty of Science, Universiti Brunei Darussalam, Gadong, Brunei Darussalam. ¹⁶⁶Plant Systematic and Ecology Laboratory, Department of Biology, Higher Teachers' Training College, University of Yaoundé I, Yaoundé, Cameroon. ¹⁶⁷Departamento de Ecologia, Universidade Federal do Rio Grande do Norte, Natal, Brazil. ¹⁶⁸Quantitative Biodiversity Dynamics, Department of Biology, Utrecht University, Utrecht, The Netherlands. ¹⁶⁹Department of Geomatics, Forest Research Institute, Raszyn, Poland. ¹⁷⁰Section for Ecoinformatics & Biodiversity, Department of Biology, Aarhus University, Aarhus, Denmark. ¹⁷¹Faculty of Forestry and Wood Sciences, Czech University of Life Sciences, Prague, Czech Republic. ¹⁷²Wildlife Conservation Society, New York, NY, USA. ¹⁷³Iwokrama International Centre for Rainforest Conservation and Development (IIC), Georgetown, Guyana. ¹⁷⁴Botanical Garden of Ural Branch of Russian Academy of Sciences, Ural State Forest Engineering University, Ekaterinburg, Russia. ¹⁷⁵Pontificia Universidad Católica del Ecuador, Quito, Ecuador. ¹⁷⁶LINCGlobal, Museo Nacional de Ciencias Naturales, CSIC, Madrid, Spain. ¹⁷⁷Plant Ecology and Nature Conservation Group, Wageningen University, P.O. Box 47 Wageningen, The Netherlands. ¹⁷⁸Silviculture Research

Institute, Vietnamese Academy of Forest Sciences, Hanoi, Vietnam. [179]Department of Biology, Stanford University, Stanford, CA, USA. [180]Centre for the Research and Technology of Agro-Environmental and Biological Sciences, CITAB, University of Trás-os-Montes and Alto Douro, UTAD, Vila Real, Portugal. [181]Department of Ecology and Sustainable Agriculture, Agricultural High School of Polytechnic Institute of Viseu, Portugal and Centre for the Research and Technology of Agro-Environmental and Biological Sciences, CITAB, University of Trás-os-Montes and Alto Douro, Vila Real, Portugal. [182]Department of Forest Engineering Universidade Regional de Blumenau, Blumenau-Santa Catarina, Brazil. [183]Environmental Studies and Research Center, University of Campinas, UNICAMP, Campinas, Brazil. [184]Department of Forest and Wood Science, University of Stellenbosch, Stellenbosch, South Africa. [185]Key Laboratory of Tropical Biological Resources, Ministry of Education, School of Life and Pharmaceutical Sciences, Hainan University, Haikou, China. [186]Division of Forestry and Natural Resources, West Virginia University, Morgantown, WV, USA. [187]Department of Zoology, University of Oxford, Oxford, UK. [188]Department of Wetland Ecology, Institute for Geography and Geoecology, Karlsruhe Institute for Technology, Karlsruhe, Germany. [189]Centre for Agricultural Research in Suriname (CELOS), Paramaribo, Suriname. [190]Tropenbos International, Wageningen, The Netherlands. [191]Polish State Forests, Coordination Center for Environmental Projects, Warsaw, Poland. [192]Research Center of Forest Management Engineering of State Forestry and Grassland Administration, Beijing Forestry University, Beijing, China. [193]INP-HB, UMRI Sciences Agronomiques et Procédés de Transformation, Yamoussoukro, Côte d'Ivoire. [194]Department of Genetics, Evolution and Environment, University College London, London, UK. ✉e-mail: irishordijk@hotmail.com

