## [Peer Review file · Nature Communications]

Effect of climate on traits of dominant and rare tree species in the world's forests

Corresponding Author: Dr Iris Hordijk

Version 0:

Reviewer comments:

Reviewer #1

(Remarks to the Author)

I would like to commend authors for their excellent work. The manuscript is very well-written, employing a relatively simple idea and method with a straightforward message. However, due to the simplicity of the analyses (which I'm not opposed to), I missed some sensitivity analyses that would add support and robustness to the methods used and the results presented. Specifically, the absence of null models and sensitivity analyses (e.g., bootstrapping and comparisons between different filters) leaves some uncertainty about the robustness of the observed patterns and their ecological interpretations. Incorporating these elements could further enhance the current findings and address potential criticisms. Below I list some major and minor issues that, in my opinion, authors could consider.

Major comments

- I am sure that authors know this dataset very well. I suspect that null models were considered, but I miss why they have not been applied. Perhaps, authors believe that both data and results are robust enough? However, my point is that some analyses and results could gain a lot and be more robust using a more robust method. For instance, authors say that some traits differ between rare and dominant species, but due to the different filters and differences in plot characteristics some of those differences could be just by chance. Also, authors say that they randomly selected 10,000 plots from the temperate biome. What if, by chance those sites are skewed towards the most distinct ones? Or even, due to the large difference in plot sizes in that biome, towards rather small or large plots? I hope not, but this can happen. Thus, I believe that some bootstrapping method and null models could tackle those issues and even improve some results/interpretations.

- In general, the methods section is well-detailed, but given the multiple methodological filters (e.g., removal of sites, selection of species based on abundance/ranking, forest age, etc.), I expected some sensitivity analyses to test the robustness of these methods and results. For instance, if authors select the top/bottom 5%, 15% or 20% species or exclude forests with more than 30, 35, 40 years. How much do the results change? How much data is lost by doing that? I'm not suggesting that sensitivity analyses should be performed for every filter, but there should be at least some analysis to demonstrate the robustness of the results and ensure that these filters are not the primary drivers of the observed outcomes.

Side note: I missed some supplemental figures showing more details about the plots used (e.g. distribution of forest age, number of species etc). These plots can help readers understand the results more deeply. For example, let's say that authors filtered plots with less than 25 years of regeneration, but most plots have ~30 years of regeneration, or around that. Is that filter really substantial? Hard to tell without knowing it.

- Different studies have used similar strategies, but most have focused on comparing functional diversity and composition across biomes globally, typically emphasizing dominant species (e.g., mass ratio effect and CWM). For example, Greenwood et al. (2017) and Bruehlheide et al. (2018) – the latter, despite methodological differences, seems to contradict the results found here, as they did not observe any effect of climatic variables on functional traits at the community level. Here, the focus was instead on mean trait values of rare and dominant species. Despite that, authors explored a bit how both groups occupy a functional space by using the PCA at the species level (Fig. S3 – still quite hard to see any pattern related to that there – perhaps some ellipses around each group would help or a density map showing how clustered rare and dominant species are?) and FDis, showing that rare species in tropical forests have a higher FDis, whereas dominant species in temperate and boreal forests have a higher FDis. Although FDis is less affected by species richness than

functional richness, it can still be. If species richness is driving FDis patterns, that approach could be used, again, under the umbrella of null models to exclude the effect of species richness in each group.

- Similarly, differences in trait values can be either inflated or diminished by the number of species in each group. While using the median can control for outliers, considering alternative approaches such as mean comparisons or using interquartile ranges might provide a more nuanced understanding of the data. Additionally, a null model could validate whether these differences are truly significant.

- Locally dominant species can be super abundant in a specific plot but absent elsewhere, indicating geographic restriction. The same can apply to rare species, with the opposite also being true for both groups when species are locally dominant or rare but geographically widespread. Although I understand the point of setting and sticking with a definition of what they consider as rare/dominant, I think it is important to address the different forms or rarity/dominance, at least in the text. Likewise, even though no hypotheses were made in that direction, knowing if most locally dominant/rare species analysed are geographically restricted or widespread can have important and different implications.

Minor comments

Kudos to authors! What a nice introduction to read!

line 42: Although I agree that splitting rare and dominant species is novel, as I suggested in the major comments, different studies have explored the effects of climate on functional diversity of species across the globe/biomes. Thus, I recommend downgrading this sentence a bit. Perhaps saying that focus on dominant species is relatively more common, but some approaches have claimed the importance of focusing on rarity as well (Violle et al. 2017)?

Line 92: citation typo (Anne E. Magurran...)

Lines 199-205: Isn't this argument likely dependent on forest age and kind of contradicts the previous discussion related to the general patterns? When discussing general patterns, authors suggest that the trait profile found might be linked to early successional species. If true, those species would be associated to more opened areas, no? Instead, here, it is suggested that species may be more adapted to shade tolerance. I know this result is relative to other environments but still. Moreover, related to the dry forests' pattern, isn't low SLA more related to water efficiency strategy? Why using a methodological argument instead of an ecological one?

Line 207: I found this paragraph too lengthy. Its core idea is well summarised at the very last sentence: gymnosperms are mostly driving this pattern. The authors could consider bringing this argument to the beginning of the paragraph and still state the importance/ecology of gymnosperms but in a more shortened way.

Line 297: this sentence looks a bit odd to me. The idea of the paragraph is to tell about the patterns along the humidity gradient, but it ends up saying that humidity is not that important. I understood that temperature is likely the major driver, but I felt the transition from here too abrupt. The previous line tells that this strategy is linked to drought resistance, but the following one tells that drought resistance is not that important. Perhaps it may help stressing, at the beginning of the paragraph, that compared to temperature humidity has a minor importance but still has some effect, which are x,y,z, and leave this "conclusion" to another paragraph.

Lines 352:355: I missed visualizing this information (e.g. forest age). Perhaps along with the boxplots, authors could make a „Withakker's-like“ figure in which both temperature and humidity are x and y axis and point size/colour could be plot size and forest age/and or species number (depending on how readable the figure will be). Just a rough idea, but I think it could help visualizing the structure of the data in a nicer way.

Lines 373:375: As I recommended in the major section, I think authors could address and potentially explore a bit more of those „different definitions of dominant and rare“ (mostly showing whether dominant and rare species are geographically restricted or widespread).

Line 392: What is the error of this imputation? The error can give a notion of how trustworthy that imputation is. If the error is high, authors can report it and write some lines in the discussion related to that. If it is low, then that's great.

Line 394: A recent paper claims that root traits comprise a different plane in the GSPFF (Carmona et al. 2023). Even though PC1 and 2 summarise ~60% of the variation, perhaps it would be worth checking whether root traits are actually more linked to PC3 and PC4 or not.

Line 440: Typo. See section 4.3, instead?

Line 480: Just a side note but I can't see the point of providing a script that it is not reproducible. I understand that data restriction can be an impediment but there are ways to tackle that issue (e.g. RData/RDS files of filtered/sample data). Perhaps this is already considered after paper's acceptance. Otherwise, better just omit the link.

References

Bruehlheide, H. et al. Global trait–environment relationships of plant communities. *Nat Ecol Evol* 2, 1906–1917 (2018).
Carmona, C. P. et al. Fine-root traits in the global spectrum of plant form and function. *Nature* 597, 683–687 (2021).
Greenwood, S. et al. Tree mortality across biomes is promoted by drought intensity, lower wood density and higher specific

leaf area. *Ecology Letters* 20, 539–553 (2017).

Violle, C. et al. Functional Rarity: The Ecology of Outliers. *Trends in Ecology & Evolution* 32, 356–367 (2017).

Reviewer #2

(Remarks to the Author)

I think this paper addresses an interesting topic, and one that I have not seen much work on. In general, the methods seem sound, though there are areas where more description is needed. I do think the writing needs substantial improvements in some places, including some minor fixes but also larger improvements to the organization and clarity.

Lines 27-46: I think this first paragraph needs substantial reworking. It addresses too wide a range of topics and with insufficient depth. One way to tighten it up would be to move topics related to climate to a new paragraph, leaving the focus of this paragraph on just species abundance distributions and functional traits.

Line 50-52: This is unclear and probably untrue. First, it is not clear what it means for a process to operate at a particular spatial scale. It may mean something like that if you want to see the small-scale abundance of a species you should consider factors like competition rather than climate. But, the precipitation that falls within a small forest plot certainly influences which species are present there, therefore having an effect on small-scale assemblages. Similarly, competition and disease frequently displace species from large sections of their potential distribution, therefore influencing abundances at the largest spatial scales. It is true that climate variables generally have more spatial autocorrelation over small spatial extents than variables like disease prevalence. But that's a very specific and different claim.

Line 152-153: This makes me wonder, given how strong the difference between angiosperms and gymnosperms is, how much do the patterns described here simply reflect gradients in dominance of those two groups (combined with the trait differences between them)? Two ways one could examine that: 1) treat angiosperms and gymnosperms as the only two taxonomic units in the study, computing mean traits and mean abundance for each across all sites 2) repeat all analyses within angiosperms and within gymnosperms.

Line 169: It is a bit weird to mention this axis as an additional trait that shows consistent differences, given that it is strongly loaded by one of the traits just mentioned. It is not really an independent result.

Line 178: It could indicate human disturbance, but also many other kinds of disturbance (e.g. windfalls).

Line 195: It is not clear why functional redundancy would lead to weaker differences between common and rare species.

Line 199: The differences in the trait values of rare species is presented here almost as a challenge to be overcome, but in the context of this study really sounds like a result. Isn't it very informative that these many rare species have highly variable traits?

Discussion: In general, I think the discussion is considerably too long. I would seek ways to simplify and compress it.

Lines 392-399: I think more detail on the trait imputation is needed. Did you, like Maynard et al. 2022, include environmental variables? It sounds from this description that, for each trait, you computed a ~20,000 plot-by-1663 species matrix (line 398, "for each of the 1663 species in each location). Is that really right? It seems unnecessary, since most species don't occur in most plots. In addition to clearer methods, I think there also needs to be more examination of the reliability of the imputation.

Line 413: Were tree ferns excluded (e.g. non-angiosperms, non-gymnosperms)?

Line 419: This part confuses me somewhat. Species are described as dominant or rare, but that only applies to a species in a site, not a species generally. There must be some species that are dominant in some places and rare in others. Why not simply compute the PCA across all species, regardless of abundance status.

Line 451: I think "humidity index" is not a very good name, since humidity refers specifically to the water content of the air. Maybe "water availability index"?

Line 69-70: Water availability is strongly related to climate, so I suggest rephrasing this a bit. Maybe just change "climate" to "temperature"?

Line 85-88: The phrasing here is somewhat awkward. At least, "leading" should be changed to "lead". But I also think it would be helpful to split this sentence into two.

Figure 2: I think the blue/purple tones are not distinct enough. I would choose a more diverse color palette. The problem is exacerbated in Figure S3, where partial transparency is used – there it is even more difficult to distinguish the shades of blue.

Figure 4: The meaning of the dashed lines is not described in the caption, but it looks like maybe it is meant to be the point at which the curve crosses the 0 line. However, in some cases it appears to be near, but not exactly at the crossing. The caption mentions confidence intervals in grey, but I cannot discern any. Each regression line appears to be two-toned, does that have something to do with the CI? Finally, displaying just the linear regression coefficient for quadratic fits is not very informative. I would show both coefficients (or I think even neither is preferable to just the one).

Version 1:

Reviewer comments:

Reviewer #1

(Remarks to the Author)

I commend the authors for the revisions made to the manuscript. The clarity of the text has improved, and the results are well supported by several sensitivity analyses. I am satisfied with the changes and have no further comments.

Reviewer #2

(Remarks to the Author)

In general, I think the authors have addressed my comments (and those of the other reviewers) well.

I still think that the color palette in Figure 2 is challenging, though I appreciate making it color blind-friendly. For me the main difficulty is in distinguishing temperate and temperate conifer, but maybe that is just my monitor, since others did not comment on it.

I may be alone in this opinion, since the other reviewers seemed to appreciate the introduction. But I still think the first paragraph could be improved. My suggestions: Remove the second sentence (dominant and rare don't really need definitions immediately), Remove the sentence defining species traits (or move it to the paragraph focused on traits later), Remove or move the sentence starting "Climate change will therefore..." since this isn't really a study of climate change. Those are my suggestions that I think could streamline this paragraph to better highlight the critical aspects of the study. But again, feel free to ignore them if you like.

The question of processes and scales of operation also appears to be one in which I am the minority. But I think language around this topic is consistently vague to the point of being misleading (not just in this paper but in the literature in general). As a result, the usage in this paper is pretty standard, but in my opinion still problematic. In my view, there are two problems. First, what does it mean for a process to operate at a particular scale? I don't think there is a clear definition of that. Second, biotic interactions like competition can be perceived at many different spatial scales. E.g. the population size of a bird within a continent may be lower because of competition with another bird species. That is because of the sum of many local-scale interactions, but competition is still important in structuring the continental-scale assemblage. In the same way, water absorption into plant roots is a fine-scale phenomenon but precipitation patterns structure large-scale patterns in assemblages.

In this specific case, I don't think a large change is required, just changing the sentence to:

"In this process, climatic factors such as temperature and precipitation, as well as biotic factors such as facilitation, competition, herbivory and pathogens act as filters on species membership in particular assemblages"
(or something like that) would work.

Overall, I would like to re-emphasize that this is a very interesting and worthwhile study. I commend the authors on their work. My criticisms at this point are minor in the overall context of the paper.

Open Access This Peer Review File is licensed under a Creative Commons Attribution 4.0 International License, which permits use, sharing, adaptation, distribution and reproduction in any medium or format, as long as you give appropriate credit to the original author(s) and the source, provide a link to the Creative Commons license, and indicate if changes were

made.

Responses to Reviewers

REVIEWER COMMENTS

Reviewer #1 (Remarks to the Author):

I would like to commend authors for their excellent work. The manuscript is very well-written, employing a relatively simple idea and method with a straightforward message.

We thank the reviewer for the time taken to review the manuscript and the positive and constructive review.

However, due to the simplicity of the analyses (which I'm not opposed to), I missed some sensitivity analyses that would add support and robustness to the methods used and the results presented. Specifically, the absence of null models and sensitivity analyses (e.g., bootstrapping and comparisons between different filters) leaves some uncertainty about the robustness of the observed patterns and their ecological interpretations. Incorporating these elements could further enhance the current findings and address potential criticisms. Below I list some major and minor issues that, in my opinion, authors could consider.

Thank you for the suggestion on performing null models and sensitivity analyses and add these analyses to the manuscript. We now include null models and sensitivity analyses to show the robustness of our models and results. Please see the detailed answers to your suggestions below.

Major comments

- I am sure that authors know this dataset very well. I suspect that null models were considered, but I miss why they have not been applied. Perhaps, authors believe that both data and results are robust enough? However, my point is that some analyses and results could gain a lot and be more robust using a more robust method. For instance, authors say that some traits differ between rare and dominant species, but due to the different filters and differences in plot characteristics some of those differences could be just by chance. Also, authors say that they randomly selected 10,000 plots from the temperate biome. What if, by chance those sites are skewed towards the most distinct ones? Or even, due to the large difference in plot sizes in that biome, towards rather small or large plots? I hope not, but this can happen. Thus, I believe that some bootstrapping method and null models could tackle those issues and even improve some results/interpretations.

These are very valid points, and we now include null models and sensitivity analyses (see also the response to your next point).

Null model of the entire dataset. In this analysis, we randomly shuffled the number of individuals per tree species within each biome. In this randomization process, the total

number of trees per plot and the total number of individuals per tree species across the specific biome was held constant. We randomized the data 100 times. Afterwards, we calculated the traits of the dominant and rare species on plot level in three different ways, using the mean, the median and the interquartile range, to test robustness of the plot level trait calculations. The null models show that calculating the plot level trait values of the dominant and rare species using interquartile ranges differs most compared to calculating the plot level trait values using the mean or median value. The interquartile range shows a peak in frequency at values close to zero for all biomes and traits, indicating that there is only one dominant or rare species identified or that trait values between dominant or rare species are very similar. We find that there is a large-to-complete overlap of frequency of trait values between dominant and rare species for the three different ways of calculating traits of dominant and rare species at plot level. The more species a biome harbours, and therefore the more random the randomization becomes, the more absolute the overlap is between the frequency of plot level dominant and rare trait values. The null model description and results can be found in the SI Fig. S4 and methods (lines (L) 416-424): *“Additionally, to verify if the results are not only caused by chance or way of calculating the dominant and rare trait values per plot, we randomised the data 100 within the five main forest biomes individually keeping the total number of trees per plot and the total number of individuals per tree species constant. Afterwards, we calculated the traits of the dominant and rare species in three different ways: using the trait mean, the median and the interquartile range. The null models indicate that, after randomization of the dataset the three different ways of calculating trait values show a large to complete overlapping frequency distribution for each of the groups, indicating robust results using the median trait plot level value (Fig S4).”*

Bootstrapping of the temperate forest biome. In the temperate forest biome, we have a total of 135,043 plots included in the database. To have a more balanced dataset, we selected 10.000 plots. Now we performed a bootstrapping analysis, in which we 100 times randomly select 10.000 plots of the temperate forest biome for all single traits. The mean and standard deviation of the dominant and rare traits resulting from the bootstrapping procedure and used in the main text are compared. The results show that the differences between dominant and rare traits we use in our main text aligns with the peak of the difference between the trait values according to the bootstrapping results. Therefore, we are confident that the difference in trait values we use are representative for the temperate broadleaf biome. The analysis description and results can be found in the SI Fig. S9. In the main text we refer to the results of the bootstrapping in (L 372-374): *“The used subset of the temperate forest biome is a good representation of the traits of dominant and rare traits in this biome, which is verified with a bootstrapping procedure (Fig. S9).”*

The effect of plot size. Regarding the effect of plot size on the species distribution, the correlations between plot size and number of dominant species ($r=0.23$, $r^2=0.05$, $p<0.01$), and plot size and number of rare species was equally weakly related ($r=0.23$, $r^2=0.05$, $p<0.01$). Additionally, the relationship between plot size and species distribution is elaborately described in Hordijk et al. (2023) & Hordijk et al. (2024). The median plot size between the different biomes is very similar as visualised in Fig. S1. Also, we considered the differences between plot sizes in our models by including plot size as an independent variable. Therefore, we argue that filtering for this specific plot size (larger then 0.02 ha and smaller then 2 ha) reduces the plot size bias and strengthen the results. We now included in

the methods (L 324-327): “See for an overview of the distribution of plot size within every forest biome Fig. S1 and the relationship between species abundances and plot size for this database Hordijk et al. (2023) and Hordijk et al. (2024).”

- In general, the methods section is well-detailed, but given the multiple methodological filters (e.g., removal of sites, selection of species based on abundance/ranking, forest age, etc.), I expected some sensitivity analyses to test the robustness of these methods and results. For instance, if authors select the top/bottom 5%, 15% or 20% species or exclude forests with more than 30,35, 40 years. How much do the results change? How much data is lost by doing that? I'm not suggesting that sensitivity analyses should be performed for every filter, but there should be at least some analysis to demonstrate the robustness of the results and ensure that these filters are not the primary drivers of the observed outcomes.

We now include an overview of the difference in dominant and rare trait values in different filter settings.

Sensitivity analyses on the definition of dominant and rare species. In this analysis we defined the dominant and rare species, as suggested by the reviewer (see below), as respectively the species with the top and bottom 5% and 15% of individuals in the plot. We decided not to incorporate the 20% definition as we think that this includes a wide range of species and not strictly the dominant and rare species. When defining dominant and rare species using the 5% definition, 17% of the plots in the database were included. When defining dominant and rare species using the 10% definition, as in the main text, 26% of the plots in the database were included. When defining dominant and rare species using the 15% definition, 27% of the database was included. For the different definitions of dominant and rare species, for each of the traits the difference in trait values between dominant and rare species was calculated. The results show that the stricter the definition of dominant and rare species (e.g. the lower the percentage), the more plots are filtered out and the more pronounced the difference in trait values between dominant and rare species becomes. The analysis description and results can be found in the SI Fig. S10. In the main text we refer to the analysis in (L 357-360): “Additionally, dominant and rare species were defined as the top and bottom 5% and 15% of individuals in the plot, showing that the stricter the definition (e.g. lower percentage), the more plots are filtered out and the more pronounced the difference between dominant and rare species trait values are (Fig. S10).”

Sensitivity analyses on filtering forest age. In this analysis we used different definitions of minimum forest age, in the main text we incorporate plots with a forest age of at least 25 years. For this analysis, as suggested by the reviewer (see below), we increased this threshold to 30 years and 35 years. We decided not to incorporate the 40 years age threshold as we think that a young secondary forest passes the first development phase well within 40 years. When defining dominant and rare species using the 30 years threshold, 83% of the database was included. When defining dominant and rare species using the 35 years threshold, 64% of the database was included. For the different age thresholds, the difference for the traits was calculated between the dominant and rare species.

The results show that there is a marginal effect of forest age when comparing differences in dominant and rare traits between 25, 30 and 35 years as the distribution in trait differences are almost identical. The analysis description and results can be found in the SI Fig. S11. In the main text we refer to the analysis in (L 318-321): “Different forest age thresholds could

potentially affect trait values of dominant and rare species. Yet, when comparing trait values using the 25-year threshold with 30- and 35-years thresholds we got very similar results (Fig. S11)." Based on these results we also deleted (L 286-292) in the discussion as filtering forest age in the range of 25 to 35 years does not have a large effect on differences between traits of dominant and rare tree species: "Especially succession is an important confounding factor, as trait differences early in succession (between light demanding pioneers and shade tolerant species regenerating in the understory) may be different than trait differences later in succession (between canopy species and understory species) (Lohbeck et al., 2014; Ruger et al., 2023). Therefore, future research should examine the link between performance of dominant and rare species, canopy strata, and succession."

Side note: I missed some supplemental figures showing more details about the plots used (e.g. distribution of forest age, number of species etc). These plots can help readers understand the results more deeply. For example, let's say that authors filtered plots with less than 25 years of regeneration, but most plots have ~30 years of regeneration, or around that. Is that filter really substantial? Hard to tell without knowing it.

This is a good suggestion. We now include figures on year of measurement, age distribution, richness and plot size (which was also included in the reviewed version) in Figure S1.

- Different studies have used similar strategies, but most have focused on comparing functional diversity and composition across biomes globally, typically emphasizing dominant species (e.g., mass ratio effect and CWM). For example, Greenwood et al. (2017) and Bruelheide et al. (2018) – the latter, despite methodological differences, seems to contradict the results found here, as they did not observe any effect of climatic variables on functional traits at the community level. Here, the focus was instead on mean trait values of rare and dominant species. Despite that, authors explored a bit how both groups occupy a functional space by using the PCA at the species level (Fig. S3 – still quite hard to see any pattern related to that there – perhaps some ellipses around each group would help or a density map showing how clustered rare and dominant species are?) and FDis, showing that rare species in tropical forests have a higher FDis, whereas dominant species in temperate and boreal forests have a higher FDis. Although FDis is less affected by species richness than functional richness, it can still be. If species richness is driving FDis patterns, that approach could be used, again, under the umbrella of null models to exclude the effect of species richness in each group.

Thank you for the suggestion to add ellipses around the different groups in Figure S3, to make clear where the trait space of the different biomes and dominant and rare species clusters are present. We have added the ellipses in this version of the manuscript (Fig. S3).

Regarding the computation of functional diversity (FDis), we have indeed computed functional diversity on biome level (Fig. S8). To compute FDis at the plot level we need at least two dominant or rare tree species per plot. However, because we are working with a global database incorporating all forest biomes we have for example in the boreal biome in the majority of the plots not two dominant and rare species. Therefore, to be able to perform the analysis including all biomes at the plot level, we decided to take the median trait values of the dominant and rare species.

We have included the recommended references in the manuscript.

- Similarly, differences in trait values can be either inflated or diminished by the number of species in each group. While using the median can control for outliers, considering alternative approaches such as mean comparisons or using interquartile ranges might provide a more nuanced understanding of the data. Additionally, a null model could validate whether these differences are truly significant.

Thank you for this suggestion, we now include an overview of differences in trait values incorporating different ways of calculating the average trait value. Please see the description of the null model and sensitivity analyses performed in the answer to the comment first major comment.

- Locally dominant species can be super abundant in a specific plot but absent elsewhere, indicating geographic restriction. The same can apply to rare species, with the opposite also being true for both groups when species are locally dominant or rare but geographically widespread. Although I understand the point of setting and sticking with a definition of what they consider as rare/dominant, I think it is important to address the different forms or rarity/dominance, at least in the text. Likewise, even though no hypotheses were made in that direction, knowing if most locally dominant/rare species analysed are geographically restricted or widespread can have important and different implications.

Thank you for this observation, it has indeed different implications if species are locally, regionally or at biome-level dominant or rare. In general, species are relatively widespread in the GFBI database, and this includes both the dominant and rare species. There is evaluated how widespread the dominant and rare species are in the GFBI database in the paper by Hordijk et al., 2024 entitled "Dominance and rarity in tree communities across the globe: Patterns, predictors and threats." Additionally, we now describe that the locally dominant and rare species are geographically relatively widespread in (L 360-362): "As described in Hordijk et al. (2024), both the dominant and rare species in the GFBI database are geographically widespread."

However, in this study we are specifically interested in the locally dominant and rare species as they are in direct contact with each other. These direct interspecific interactions can result in for example facilitation or competition, which could result in either similar or dissimilar trait values. The reasoning behind opting for locally dominant and rare species is explained in (L 345-348): "We defined dominant and rare species at the plot level, as this is the spatial scale at which species interact more directly with each other, and therefore the outcome of both abiotic and biotic interactions affecting species abundances are reflected (Roughgarden & J., 1986; Stroud et al., 2015)." We have checked the manuscript on consistent wording regarding the emphasis on locally dominant and rare species and made adjustments when needed.

Minor comments

Kudos to authors! What a nice introduction to read!

Thank you very much for your positive words.

line 42: Although I agree that splitting rare and dominant species is novel, as I suggested in the major comments, different studies have explored the effects of climate on functional diversity of species across the globe/biomes. Thus, I recommend downgrading this sentence a bit. Perhaps saying that focus on dominant species is relatively more common, but some approaches have claimed the importance of focusing on rarity as well (Violle et al. 2017)?

Thank you for this suggestion. We have downgraded the sentence and changed it to (L 42-44): *“However, the extent to which individual traits of locally dominant and rare tree species differ, and how these differences are affected by climate, remains largely unexplored at a global scale.”*

Line 92: citation typo (Anne E. Magurran...)

The citation is corrected.

Lines 199-205: Isn't this argument likely dependent on forest age and kind of contradicts the previous discussion related to the general patterns? When discussing general patterns, authors suggest that the trait profile found might be linked to early successional species. If true, those species would be associated to more opened areas, no? Instead, here, it is suggested that species may be more adapted to shade tolerance. I know this result is relative to other environments but still.

If traits patterns are linked to earlier successional phases the traits are in general expected to show traits more related to light demanding pioneers compared to shade tolerant species (Lohbeck et al., 2014; Rüger et al., 2023). The general patterns in trait differences indeed suggest that the dominant species are earlier successional, while the rare species are later successional species (L 178-182). This pattern is confirmed by the tropical biomes where dominant species express a lower SLA compared to rare species. We realized that in this sentence (L 204-208) a word was missing. We now changed this to, with changes in bold to: *“Dominant tropical forest species had, next to the three traits mentioned above, also a lower SLA (Fig S6), which may reflect stronger adaptations to shade **for rare species**, as moist tropical forests tend to be more dense and continuously shaded compared to other forest biomes (Reich et al., 2003; Valladares & Niinemets, 2008).”*

Moreover, related to the dry forests' pattern, isn't low SLA more related to water efficiency strategy? Why using a methodological argument instead of an ecological one?

In dry tropical forests, a high SLA indicates a drought-avoidance strategy of drought deciduous species while a low SLA indicates a drought-tolerance strategy of evergreen species (Ramírez-Valiente & Cavender-Bares, 2017). In the dominant canopy species, we would indeed expect a drought-tolerance strategy and a lower SLA (Legner et al., 2014). Therefore, we changed the sentence to (L 208-210): *“Regarding the dry tropical forests, a lower SLA could indicate a drought-tolerance strategy of the dominant, sun exposed canopy trees (Legner et al., 2014; Ramírez-Valiente & Cavender-Bares, 2017).”*

Legner, N., Fleck, S., & Leuschner, C. (2014). Within-canopy variation in photosynthetic capacity, SLA and foliar N in temperate broad-leaved trees with contrasting shade tolerance. *Trees*, 28, 263-280.

Ramírez-Valiente, J. A., & Cavender-Bares, J. (2017). Evolutionary trade-offs between drought resistance mechanisms across a precipitation gradient in a seasonally dry tropical oak (*Quercus oleoides*). *Tree physiology*, 37(7), 889-901.

Line 207: I found this paragraph too lengthy. Its core idea is well summarised at the very last sentence: gymnosperms are mostly driving this pattern. The authors could consider bringing this argument to the beginning of the paragraph and still state the importance/ecology of gymnosperms but in a more shortened way.

Thank you for this suggestion. As also suggested by another reviewer, the discussion is quite lengthy, and it is advised to focus on the most important results to discuss. Originally, the paragraph occupied line 208 to 236 and is now shortened from line 212 to 229.

Line 297: this sentence looks a bit odd to me. The idea of the paragraph is to tell about the patterns along the humidity gradient, but it ends up saying that humidity is not that important. I understood that temperature is likely the major driver, but I felt the transition from here too abrupt. The previous line tells that this strategy is linked to drought resistance, but the following one tells that drought resistance is not that important. Perhaps it may help stressing, at the beginning of the paragraph, that compared to temperature humidity has a minor importance but still has some effect, which are x,y,z, and leave this "conclusion" to another paragraph.

We moved the conclusion which was placed under the paragraph explaining water availability-trait patterns to the start of the paragraph where we start discussing the effect of climate on traits (L 231).

Lines 352:355: I missed visualizing this information (e.g. forest age). Perhaps along with the boxplots, authors could make a „Withakker’s-like“ figure in which both temperature and humidity are x and y axis and point size/colour could be plot size and forest age/and or species number (depending on how readable the figure will be). Just a rough idea, but I think it could help visualizing the structure of the data in a nicer way.

Thank you for this suggestion. We now include figures on year of measurement, age distribution, richness and plot size (which was also included in an earlier version) in Figure S1.

Lines 373:375: As I recommended in the major section, I think authors could address and potentially explore a bit more of those „different definitions of dominant and rare“ (mostly showing whether dominant and rare species are geographically restricted or widespread).

There is evaluated how widespread the dominant and rare species in the GFBI database are in the paper Hordijk et al., 2024 titled "Dominance and rarity in tree communities across the globe: Patterns, predictors and threats." Additionally, we now describe that the locally dominant and rare species are geographically relatively widespread in (L 360-362): "As described in Hordijk et al. (2024), both the dominant and rare species in the GFBI database are geographically widespread."

Line 392: What is the error of this imputation? The error can give a notion of how trustworthy that imputation is. If the error is high, authors can report it and write some lines in the discussion related to that. If it is low, then that's great.

This is a good point, and we now include a discussion of the role of trait uncertainty in the text (L 281-285): *“The trait values used here were estimated based on phylogenetic and environmental information. This allowed for the incorporation of trait plasticity across environmental gradients, but it also introduces model-based uncertainty into the predictions. The imputation uncertainty has been shown to have negligible bias when averaging over many species (Maynard et al. 2022).”* Indeed, this question was a central focus of Maynard et al. For the nine traits considered here, the accuracy of the trait estimated models ranged from $R^2=0.41$ (root depth) to $R^2=0.69$ (seed dry mass), with an average of $R^2=0.57$ across all traits, showing consistent prediction accuracy. Moreover, they show that these imputed traits yield almost identical patterns as with the raw data (Fig S20-S21 in Maynard et al), and that these results are robust to the degree of imputation in the data (Fig S15), highlighting that the imputation errors introduce negligible bias. Nevertheless, they also caution against using the non-logged values, as the errors are log-normally distributed and thus are proportional to the mean. This is why we use logged values here (L 394), to maximize the stability of the predicted values.

We hope the discussion of this in the text helps clarify this issue, and we are happy to include additional information if the reviewer has any suggestions.

Line 394: A recent paper claims that root traits comprise a different plane in the GSPFF (Carmona et al. 2023). Even though PC1 and 2 summarise ~60% of the variation, perhaps it would be worth checking whether root traits are actually more linked to PC3 and PC4 or not.

Thank you for this comment, we have analysed if the root traits do relate to PC3 or PC4. PC3 is mainly related to wood density, bark thickness, water availability and temperature. PC4 is mainly related to height, crown diameter and root depth. Therefore, PC4 is indeed related to root depth, but as this is the only root trait considered in this study I do not think that we are able to classify PC4 as a root trait plane. Additionally, as PC4 is also related to height and crown diameter this plane could be classified as a tree architectural plane.

Line 440: Typo. See section 4.3, instead?

Thank you for observing this error, it should indeed be section 4.3 instead.

Line 480: Just a side note but I can't see the point of providing a script that it is not reproducible. I understand that data restriction can be an impediment but there are ways to tackle that issue (e.g. RData/RDS files of filtered/sample data). Perhaps this is already considered after paper's acceptance. Otherwise, better just omit the link.

Once the manuscript is published the data and code to recreate the main graphs will be made available.

References

- Bruehlheide, H. et al. Global trait–environment relationships of plant communities. *Nat Ecol Evol* 2, 1906–1917 (2018).
- Carmona, C. P. et al. Fine-root traits in the global spectrum of plant form and function. *Nature* 597, 683–687 (2021).

Greenwood, S. et al. Tree mortality across biomes is promoted by drought intensity, lower wood density and higher specific leaf area. *Ecology Letters* 20, 539–553 (2017).
Violle, C. et al. Functional Rarity: The Ecology of Outliers. *Trends in Ecology & Evolution* 32, 356–367 (2017).

Reviewer #2 (Remarks to the Author):

I think this paper addresses an interesting topic, and one that I have not seen much work on. In general, the methods seem sound, though there are areas where more description is needed. I do think the writing needs substantial improvements in some places, including some minor fixes but also larger improvements to the organization and clarity.

Thank you for taking the time and effort to review this manuscript. We answer to your specific comments in detail below.

Lines 27-46: I think this first paragraph needs substantial reworking. It addresses too wide a range of topics and with insufficient depth. One way to tighten it up would be to move topics related to climate to a new paragraph, leaving the focus of this paragraph on just species abundance distributions and functional traits.

The goal of this first paragraph in the introduction is to introduce the most important topics and to set the stage for the rest of the introduction and manuscript. As climate is, together with traits of dominant and rare species, the main topic of the manuscript, we think that it is worthwhile to include the topic climate in the first paragraph of the introduction. In the fourth paragraph of the introduction (starting at L 70), we discuss in detail the effect of climate on traits.

Line 50-52: This is unclear and probably untrue. First, it is not clear what it means for a process to operate at a particular spatial scale. It may mean something like that if you want to the small-scale abundance of a species you should consider factors like competition rather than climate. But, the precipitation that falls within a small forest plot certainly influences which species are present there, therefore having an effect on small-scale assemblages. Similarly, competition and disease frequently displace species from large sections of their potential distribution, therefore influencing abundances at the largest spatial scales. It is true that climate variables generally have more spatial autocorrelation over small spatial extents than variables like disease prevalence. But that's a very specific and different claim.

In this section we try to make the claim that global temperature and precipitation patterns are influenced by processes at a large spatial scale, while direct interactions with neighbouring plants are processes which take place at a local scale. Temperature and precipitation can also be more homogenous over large areas, while neighbourhood interactions are restricted to the local environment. We do agree that large-scale processes affect local conditions, and we also argue that there are processes which take place at different spatial scales.

Line 152-153: This makes me wonder, given how strong the difference between

angiosperms and gymnosperms is, how much do the patterns described here simply reflect gradients in dominance of those two groups (combined with the trait differences between them)? Two ways one could examine that: 1) treat angiosperms and gymnosperms as the only two taxonomic units in the study, computing mean traits and mean abundance for each across all sites 2) repeat all analyses within angiosperms and within gymnosperms.

Thank you for this comment, we explored this question in detail. To do so, we added the gymnosperm arrow in the PCA (Fig. 2) and evaluated how the loadings on the first PC axis (Fig. 3) and percentage gymnosperms differ between dominant and rare species (Fig. S6). The PCA shows that the differences in trait values along the first PC axis is mainly explained by the division between gymnosperms and angiosperms (Fig. 2). Additionally, the loadings on the first PC axis are higher for the dominant species compared to the rare species across all biomes (Fig. 3). However, when evaluating if the percentage gymnosperms belonging to dominant or rare species differ, we found that only for the temperate and temperate conifer forest the percentage gymnosperms is higher for the dominant species (Fig. S6). Therefore, we conclude that the division between gymnosperms and angiosperms is clear in terms of trait differences (in line with e.g. Maynard et al., 2022), but that the gymnosperms do not primarily drive the differences between traits in dominant and rare species.

Line 169: It is a bit weird to mention this axis as an additional trait that shows consistent differences, given that it is strongly loaded by one of the traits just mentioned. It is not really an independent result.

The PC axes are indeed a summary of different traits, as is the division between gymnosperms and angiosperms. We now make this clear in the wording across the manuscript and do refer to 13 traits as “11 traits and 2 multivariate trait axes”. We made these changes in L 14, 102, 169 & 191. Regarding the independence of the results between the individual traits and the gymnosperm/angiosperm division and PC axes, the latter are indeed multivariate trait axes rather than independent trait values. However, we think that it has added value to both present the trait axes and individual traits in the results. With the traits axes overall patterns can be detected, while with the individual traits these patterns can be explained.

Line 178: It could indicate human disturbance, but also many other kinds of disturbance (e.g. windfalls).

That is a good point, a disturbance could also be a natural disturbance. We changed the sentence to (L 179): “...a possible indication of human or natural disturbance in the forests evaluated in this study (Li et al., 2023; R uger et al., 2023).”

Line 195: It is not clear why functional redundancy would lead to weaker differences between common and rare species.

We now elaborate on the underlying logic and rewrote the sentence to (L 200-202): “A higher functional redundancy may result in less striking trait differences between dominant and rare species as species have more similar trait values (Dalerum et al., 2012).”

Line 199: The differences in the trait values of rare species is presented here almost as a challenge to be overcome, but in the context of this study really sounds like a result. Isn't it very informative that these many rare species have highly variable traits?

The trait space of dominant and rare species is not the focus of this study, but the trait range of dominant and rare species is indeed informative. The reason that it is presented here as a challenge is that we calculate the median trait value of dominant and rare species. Ideally, this median does reflect the average trait of the dominant and rare species well. However, when it reflects better the average trait of the dominant species compared to the rare species, this can introduce a bias in the analyses.

Discussion: In general, I think the discussion is considerably too long. I would seek ways to simplify and compress it.

Thank you for this suggestion. We have shortened the following sections of the discussion:

- Paragraph on gymnosperms and angiosperms occupied line 208 to 236 and is now shortened from line 212 to 229.
- Paragraph on differences between biomes on crown diameter, leaf nitrogen, and seed mass of dominant and rare species is erased (270 words).

Lines 392-399: I think more detail on the trait imputation is needed. Did you, like Maynard et al. 2022, include environmental variables? It sounds from this description that, for each trait, you computed a ~20,000 plot-by-1663 species matrix (line 398, "for each of the 1663 species in each location). Is that really right? It seems unnecessary, since most species don't occur in most plots. In addition to clearer methods, I think there also needs to be more examination of the reliability of the imputation.

That part of the methods was not clearly described indeed. We changed the sentence to (L 381-384): "*These models incorporate intraspecific variation, and thus provide a unique prediction of each trait for each of the 1,663 species in each location where the species occurs, based on the combination of phylogenetic and environmental information.*" That is, rather than make a 20,000 x 1663 prediction matrix for all species at every location, we only predict the trait expression for each tree species in the locations where it occurs (not all 20,000 plots), thus giving a unique set of traits for each species, depending on its phylogenetic information and the local environment. The resulting matrix thus has a row for each species for each unique geographic location where that species occurs, and the columns represent the predicted traits of that species at that location.

We hope this clarification in the text addresses this issue, and please also see our response to Reviewer 1 regarding the imputation accuracy.

Line 413: Were tree ferns excluded (e.g. non-angiosperms, non-gymnosperms)?

Tree ferns and palms are not included in the analyses, which we now make clear in L 306-307: "*Tree ferns and palms are not included in the database.*"

Line 419: This part confuses me somewhat. Species are described as dominant or rare, but

that only applies to a species in a site, not a species generally. There must be some species that are dominant in some places and rare in others. Why not simply compute the PCA across all species, regardless of abundance status.

Thank you for this suggestion. The aim of this study is to analyse the species which are locally dominant or rare. However, we also explored the PCA including all dominant and rare species with average trait values per species (Fig. S3). The overall pattern in the PCA is similar to the pattern observed when the intraspecific trait variation is considered (Fig. 2). The main difference is that the arrows of temperature and humidity are aligned in the PCA incorporating the average trait values, while in the PCA with intraspecific trait variation temperature and humidity show arrows in almost opposite directions. This might be explained by the proportionally larger number of species in the tropical forest in the PCA with average trait values per species. In the tropical moist forest, the environment is more humid with a higher temperature, and this is also the direction the arrows point towards. In the PCA including intraspecific trait variation the number of dominant and rare species is more balanced between the biomes and temperature and humidity are more independent.

Line 451: I think “humidity index” is not a very good name, since humidity refers specifically to the water content of the air. Maybe “water availability index”?

The “humidity index” is climatic water availability expressed as the ratio of mean annual precipitation over mean annual evapotranspiration. To not cause confusion on the terminology, we changed “humidity index” to “water availability” in the manuscript and the figures.

Line 69-70: Water availability is strongly related to climate, so I suggest rephrasing this a bit. Maybe just change “climate” to “temperature”?

Thank you for the suggestion, “climate” is changed to “temperature”.

Line 85-88: The phrasing here is somewhat awkward. At least, “leading” should be changed to “lead”. But I also think it would be helpful to split this sentence into two.

We rephrased the sentence as follows (L 86-89): “*Extreme temperatures in combination with drought can exacerbate water stress, damage plant tissues, and ultimately lead to plant mortality and species exclusion (Lintunen et al., 2013; Pollastrini et al., 2019; Ruehr et al., 2015).*”

Figure 2: I think the blue/purple tones are not distinct enough. I would choose a more diverse color palette. The problem is exacerbated in Figure S3, where partial transparency is used – there it is even more difficult to distinguish the shades of blue.

Thank you for this observation. We have now added ellipses around the trait values of the dominant and rare species per biome in Figure S3 to increase readability. The colour scheme we are using is tested and appropriate for three different types of colour blindness (deuteranopia, protanopia and tritanopia). Additionally, overall readability and distinctiveness of the colour scheme is confirmed by the main co-authors. However, if the editor feels very strongly it is no problem at all to change the colour scheme across all figures.

Figure 4: The meaning of the dashed lines is not described in the caption, but it looks like maybe it is meant to be the point at which the curve crosses the 0 line. However, in some cases it appears to be near, but not exactly at the crossing.

The dashed line is placed now more accurately at the crossing with the zero line.

The caption mentions confidence intervals in grey, but I cannot discern any. Each regression line appears to be two-toned, does that have something to do with the CI?

The two-toned graph is indeed a result of the 95% confidence interval line. As the confidence interval is not so visible indeed, we removed it.

Finally, displaying just the linear regression coefficient for quadratic fits is not very informative. I would show both coefficients (or I think even neither is preferable to just the one).

Thank you for this suggestion, we now added both regression coefficients to the figure.

Responses to Reviewers

REVIEWERS' COMMENTS

Reviewer #1 (Remarks to the Author):

I commend the authors for the revisions made to the manuscript. The clarity of the text has improved, and the results are well supported by several sensitivity analyses. I am satisfied with the changes and have no further comments.

We thank the reviewer to take the time to review the manuscript and we are glad that the revisions and adaptation have been approved.

Reviewer #2 (Remarks to the Author):

In general, I think the authors have addressed my comments (and those of the other reviewers) well.

We thank the reviewer to take the time to review the manuscript and we are glad that the revisions and adaptation are acknowledged and approved.

I still think that the color palette in Figure 2 is challenging, though I appreciate making it color blind-friendly. For me the main difficulty is in distinguishing temperate and temperate conifer, but maybe that is just my monitor, since others did not comment on it.

Thank you for highlighting this observation. Overall readability and distinctiveness of the colour scheme is confirmed by the main co-authors. However, if the editor feels very strongly it is no problem at all to change the colour scheme across all figures.

I may be alone in this opinion, since the other reviewers seemed to appreciate the introduction. But I still think the first paragraph could be improved. My suggestions: Remove the second sentence (dominant and rare don't really need definitions immediately), Remove the sentence defining species traits (or move it to the paragraph focused on traits later), Remove or move the sentence starting "Climate change will therefore..." since this isn't really a study of climate change. Those are my suggestions that I think could streamline this paragraph to better highlight the critical aspects of the study. But again, feel free to ignore them if you like.

Thank you very much for your suggestions. The first paragraph of the introduction is changed according to your feedback. Specifically, we removed lines(L) 28-30: “Dominant and rare species are respectively the most and least abundant species in the community and abundance is defined in this study as number of individuals in the local community.” Additionally, L31-33 is moved to L57-58: “A trait is defined as any morphological, physiological or phenological feature measurable at the individual plant level that affects plant performance (Violle et al., 2007).” L36-38 Highlights the relevance of this study in a larger context: “Climate change will therefore have a strong effect on the occurrence and distribution of forest biomes, traits, and consequently, forest ecosystem functioning (Joswig et al., 2022; Kühn et al., 2021; Madani et al., 2018).” Therefore, this sentence is kept in the first paragraph of the introduction. However, if the editor feels strongly about this sentence we can remove or move it to another section of the introduction.

The question of processes and scales of operation also appears to be one in which I am the minority. But I think language around this topic is consistently vague to the point of being misleading (not just in this paper but in the literature in general). As a result, the usage in this paper is pretty standard, but in my opinion still problematic. In my view, there are two problems. First, what does it mean for a process to operate at a particular scale? I don't think there is a clear definition of that. Second, biotic interactions like competition can be perceived at many different spatial scales. E.g. the population size of a bird within a continent may be lower because of competition with another bird species. That is because of the sum of many local-scale interactions, but competition is still important in structuring the continental-scale assemblage. In the same way, water absorption into plant roots is a fine-scale phenomenon but precipitation patterns structure large-scale patterns in assemblages.

In this specific case, I don't think a large change is required, just changing the sentence to: “In this process, climatic factors such as temperature and precipitation, as well as biotic factors such as facilitation, competition, herbivory and pathogens act as filters on species membership in particular assemblages” (or something like that) would work.

Thank you very much for your suggestion, we changed the sentence accordingly to L47-50: “In this process, climatic factors such as temperature and precipitation, as well as biotic factors such as facilitation, competition, herbivory and pathogens act as filters on species membership in particular assemblages (Bruehlheide et al., 2018; Weiher & Keddy, 2001).”

Overall, I would like to re-emphasize that this is a very interesting and worthwhile study. I commend the authors on their work. My criticisms at this point are minor in the overall context of the paper.

Thank you very much for your positive words regarding our work.